# About the Accuracy and Problems of Consumer Devices in the Assessment of Sleep

**DOI:** 10.3390/s19194160

**Published:** 2019-09-25

**Authors:** Mohamed S. Ameen, Lok Man Cheung, Theresa Hauser, Michael A. Hahn, Manuel Schabus

**Affiliations:** 1Laboratory for Sleep, Cognition and Consciousness Research, Department of Psychology, University of Salzburg, Hellbrunner Strasse 34, 5020 Salzburg, Austria; 2School of Psychology, University of Surrey, Stag Hill, Guildford GU2 7XH, UK; 3Center for Cognitive Neuroscience Salzburg (CCNS), Hellbrunner Strasse 34, 5020 Salzburg, Austria

**Keywords:** wrist-worn devices, sleep trackers, activity trackers, sleep classification, polysomnography

## Abstract

Commercial sleep devices and mobile-phone applications for scoring sleep are gaining ground. In order to provide reliable information about the quantity and/or quality of sleep, their performance needs to be assessed against the current gold standard, i.e., polysomnography (PSG; measuring brain, eye, and muscle activity). Here, we assessed some commercially available sleep trackers, namely an activity tracker; Mi band (Xiaomi, Beijing, China), a scientific actigraph: Motionwatch 8 (CamNTech, Cambridge, UK), and a much-used mobile phone application: Sleep Cycle (Northcube, Gothenburg, Sweden). We recorded 27 nights in healthy sleepers using PSG and these devices and compared the results. Surprisingly, all devices had poor agreement with the PSG gold standard. Sleep parameter comparisons revealed that, specifically, the Mi band and the Sleep Cycle application had difficulties in detecting wake periods which negatively affected their total sleep time and sleep-efficiency estimations. However, all 3 devices were good in detecting the most basic parameter, the actual time in bed. In summary, our results suggest that, to date, the available sleep trackers do not provide meaningful sleep analysis but may be interesting for simply tracking time in bed. A much closer interaction with the scientific field seems necessary if reliable information shall be derived from such devices in the future.

## 1. Introduction

Our knowledge about the structure and function of sleep is derived mainly from recordings that are done in sleep laboratories. In these recordings, physiological activity is measured using polysomnography (PSG), which is considered as the gold standard to assess sleep. PSG requires a combination of electroencephalography (EEG), electrooculography (EOG), and electromyography (EMG) data. Although these recordings contribute widely to our constantly expanding knowledge about sleep, their major drawback is that they do not mimic the habitual sleeping environment at home. The laboratory setup, experimental manipulations, and the first night effect, among other factors, form challenges that hinder laboratory sleep recordings from properly reflecting at-home sleep [1,2,3]. For instance, time in bed (TiB), total sleep time (TST), as well as sleep efficiency (SE) are significantly reduced during a night in the laboratory as compared to a night at home [1,2]. Therefore, laboratory sleep recordings might not be optimal in measuring sleep and consequently in diagnosing sleep disorders. Indeed, home-based PSG recordings show better dissociation between healthy sleepers and insomniacs than laboratory PSG [4]. Altogether, it is a priority in the field of sleep research and sleep medicine to develop better tools that can accurately and reliably measure sleep at home and in a wider range of population.

In that context, we have recently witnessed a vast increase in the available consumer devices, i.e., sleep trackers and mobile-phone applications, which aim to assess and ultimately improve sleep. These devices have the potential to overcome the bias induced by laboratory settings as they are designed to assess sleep outside the laboratory with minimal effort from the user. Before this is realized, it is essential to scientifically test and validate these devices against the “gold standard” to ensure that such devices and applications do not provide inaccurate information to the naïve end-user, resulting in “unintended effects on sleep beliefs and behaviors” [5]. The adherence of such devices and applications to the gold standard ensures that their reliability and validity upholds the ethical standards before they are advertised by the industry.

In that regard, previous studies have assessed the performance of a number of sleep trackers in measuring sleep [6,7,8]. Accelerometer-based sleep trackers, for instance, had high accuracy in detecting sleep but low accuracy in detecting awakenings from sleep which makes them quite efficient in estimating parameters such as TST and/or TiB but not that accurate in estimating wake after sleep onset (WASO) [6]. Similarly, mobile-phone applications had very poor agreement with the PSG gold standard in estimating sleep parameters such as SE and sleep onset latency (SOL) as well as in staging sleep into light and deep sleep [7]. In such a rapidly evolving market, however, it is essential to regularly update our knowledge about the current status of such devices and applications in order to constantly monitor their progress and optimize their performance. Therefore, the aim of this study was to assess the performance of some of these readily used consumer devices which claim to monitor sleep and to provide reliable information about sleep quality and sleep architecture on a nightly basis. Specifically, we assessed sleep data from two devices: (1) a commercial activity tracker, the Mi band (MB; Xiaomi, Beijing, China), and (2) a scientific actigraphy, Motionwatch 8 (MW; CamNTech, Cambridge, UK), as well as one readily used mobile-phone application: the Sleep Cycle (SC; Northcube, Gothenburg, Sweden). Our aim was to select some of the relatively cheap and highly popular commercial sleep trackers, the Mi band and the Sleep Cycle application, as well as at least one research-grade sleep tracker, the Motionwatch to ensure that we tested at least one device/application from each class of sleep trackers available. We compared sleep parameters as measured by these trackers against our PSG gold standard that relied on semi-automatic sleep staging using the SOMNOlyzer 24X7 solution [9,10].

## 2. Materials and Methods

Study sample: For the study, we recorded a total of 27 nights of PSG recordings. First, we recruited 19 healthy participants (13 females, mean age: 29 ± 13. Range: 19–64 years) for in-laboratory recordings. Participants arrived at the sleep laboratory of the University of Salzburg at 9 pm. They were instructed about the procedure and the purpose of the experiment. After signing the consent forms, they were given the Mi band (MB) and the MotionWatch (MW). After we confirmed that both the MB and the MW were recording, we started with PSG preparation. Before turning the lights off and starting the PSG recording, we started the Sleep Cycle application (SC) and placed the device next to the subject. Participants went to bed at around 11 pm and stayed in bed (TIB, time in bed) for approximately 8 h (452.29 ± 81.78 min). Two of these participants had to be excluded from our analyses due to technical problems with the PSG recordings. Additionally, we recorded 8 ambulatory “home” PSG nights using an ambulatory EEG device together with the MB device and the SC application after the participants visited the lab for electrode placement. Due to technical problems with some of the devices/application, we finally analyzed 21 nights for MB and 12 nights for MW and SC. 

Instruments: Mi Band (MB; Xiaomi, Beijing, China) is a wrist-worn commercial activity tracker (15.7 mm × 10.5 mm × 40.3mm and weighs 7.0 g) that detects sleep using a combination of two sensors: (i) a proximity sensor and (ii) actigraphy (actimetric sensor). While the proximity sensor detects contact with the skin, the actimetric sensor detects body movements to count sleep time and to differentiate between light and deep sleep. MB uses a triaxial accelerometer and a photoplethysmography (PPG) sensor to detect movements and to monitor blood-volume changes, respectively. MB starts recording sleep when it detects no movement for a certain (unspecified) period of time. MB classifies sleep into deep and light sleep based on body movements. In addition, MB uses PPG for continuous heart-rate monitoring during sleep to track light and deep sleep more precisely. However, information on the thresholds used for classifying sleep is not made available. The collected data are then converted to a hypnogram using the Mi Fit software (algorithm v1.1.14, Anhui Humai Information Technology CO., Ltd., Hefei, China). Generally, we used the Mi band 2; however, some of the recordings were done using the Mi band 3, which, as advised by Xiaomi developers on their website, has no hardware update from the Mi band 2 that might influence the results. For our analysis, we used the hypnogram (graph) produced by the Mi Fit software to extract the exact timings of the start and the end of sleep, the start and the end of each sleep stage (light sleep/deep sleep), as well as the duration of WASO in minutes. Using these data, we calculated the TiB as the difference between the start and the end times reported in the software and the TST as the total amount of time spent asleep (both light and deep sleep) during this period. SOL is defined as the difference between the start time of the PSG recording and the time when MB starts recording light sleep. Finally, SE was calculated as follows; SE = (TST/TiB) × 100. The 30-second epochs for the epoch-by-epoch agreement were extracted manually by fragmenting the segments of wake/light sleep/deep sleep in the graph into 30-s segments.

MotionWatch 8 (MW; CamNTech, Cambridge, UK) is a research-grade actigraphy with a built-in ambient light sensor (Dimensions: 36-mm length × 28.2-mm width × 9.4-mm depth excluding strap and weighs 9.1 g including the battery but excluding the strap; sensor: triaxial accelerometer, microelectromechanical (MEM) technology, 0.01 g to 8 g range, 3–11 Hz). For scoring sleep, MW uses a triaxial microelectromechanical system (MEMS) accelerometer (range: 0.01 g–8 g, bandwidth: 3–11Hz) to monitor body movements with a sampling frequency of 50Hz. The onboard software processes the raw acceleration data such that one quantitative measure of activity for a predefined epoch length of, e.g., 30 s is calculated and stored on an internal nonvolatile memory. For differentiating between sleep and wake epochs, we used the MotionWare software (v1.1.20, empire Software GmbH, Cologne, Germany) which uses an algorithm that depends on thresholding. Briefly, this thresholding algorithm assigns an activity score to each epoch by totalizing the epoch in question and those surrounding it using weighting factors. If the activity score of an epoch is above a predefined threshold, then the epoch is scored as wake; otherwise, it is scored as sleep. This activity score is dependent upon the sampling epoch-length. Since we used 30-s epochs, the activity scores are calculated as follows: the number of movements in the epoch being scored is multiplied by 2 and that of the epochs within 2 minutes of this epoch (1 minute before and 1 minute after) are multiplied by 0.2. The activity score of this epoch is the sum of these weighted values. MW has 3 thresholding options: low, medium, and high corresponding to 80, 40, and 20 activity scores, respectively. For our analysis, we used the high-threshold value as it was used for the validation of MW. Moreover, an additional level of scoring is also done based on the movements detected by MW per epoch. If there are more than 2 movements in a 30-s epoch, this epoch is scored as mobile (awake). The detection of the start of sleep is based on 10-minute segments and is totally independent of the sleep/wake scoring described earlier as the threshold is 3 activity counts in a 30-s epoch. The process starts by looking at the first 10 minutes (twenty 30-s epochs) after lights-out (which was synchronized to the PSG). Each epoch is tested against the threshold (i.e., 3 counts), and the number of epochs exceeding the threshold is counted. If this number is greater than 2, then the process is repeated 1 minute later. This process continues until a 10-minute block that fulfills the criteria is detected, marking the start of sleep. Detecting the end of sleep is done using the same procedure; however, instead of 10-minute segments of maximum 2 epochs containing 3 or more activity counts, 5-minute segments of maximum 2 epochs containing 5 or more activity counts marks the end of sleep. Note that lights-out was always synchronized to the start of the PSG recording. TiB, TST, SOL, and SE are calculated automatically by the software, while WASO is calculated manually. TiB is defined as the total elapsed time between the “Lights Out” and “Got Up” times. TST is defined as the total elapsed time between the “Fell Asleep” and “Woke Up” times. SOL is defined as the time between “Lights Out” and “Fell Asleep”, and SE is the “Actual Sleep Time” divided by “Time in Bed” in percentage. Finally, WASO is determined by calculating the number of epochs scored as awake after the first epoch of sleep. 30-second epoch data provided by the software is used to calculate the epoch-by-epoch agreement with the gold standard.

Sleep Cycle (SC; NorthCube, Gothenburg, Sweden) is a mobile-phone application that is available on android-based as well as iOS-based devices. SC is a smart alarm-clock that tracks your sleep patterns and wakes you up during light sleep. SC tracks sleep throughout the night and use a 30-minute window that ends up with the desired alarm time during which the alarm goes off at the lightest possible sleep stage (i.e., light sleep). SC scores sleep through motion detection via one of two motion-detection modes: (i) microphone, which uses the built-in microphone to analyze movements, or (ii) accelerometer, which uses the phone’s built-in accelerometer. SC tracks movements through the night and uses them to detect and score sleep as well as to plot a graph (hypnogram). For our analysis, we used SC on both iOS (v5.7.1) and Android (v3.0.1). We were advised by the developers (Northcube, Gothenburg, Sweden) that there is no difference between the two versions in the sleep-scoring algorithm. We used the recommended settings for recording; that is, we used the microphone to detect movements and we placed the phone next to the participant, with the microphone facing the participant and the charger plugged in. By selecting the microphone option to monitor sleep, the SC application uses sound analysis to identify sleep phases by tracking movements in bed. The SC application uses the smartphone’s built-in microphone to pick up sounds from the sleeper. After receiving the sound input, the application then filters the sound using a series of high and low cut-off filters to identify specific noises that correlate with movement. When there is no motion, the application registers deep sleep; when there is little motion, it registers light sleep; and when there is a lot of motion, it registers wakefulness. More details on the algorithms and the technical aspects of sound analysis are not available to the public. Although we used the premium version, we were not able to find any information on the algorithm SC uses for sleep scoring. As even the premium version of the application provides no access to the raw data, we performed image analysis of the hypnogram provided by the application (see the Appendix A for more information) as a workaround. Through this method, we were able to extract 30-s-epoch information about sleep scoring (wake/light sleep/deep sleep) which was also used for measuring sleep parameters. TiB was calculated based on the “went to sleep” and waking times reported by the application. TST was calculated by subtracting all the “awake” epochs from the TiB. SOL was calculated by summing up all the awake 30-s epochs before the first light-sleep epoch. WASO was calculated by summing up all the awake epochs that lie between the first light-sleep epoch and the “woke up” time. SE = (TST/TiB) × 100. 

EEG data acquisition: For the nights spent in the laboratory, brain activity was recorded using high-density EEG with a 256-electrode GSN HydroCel Geodesic Sensor Net (Electrical 478 Geodesics Inc., Eugene, Oregon, USA) and a Net Amps 400 amplifier. Additionally, we recorded electrocardiography (ECG), electromyography (EMG), and electrooculography (EOG) using bipolar electrodes. Ambulatory PSG was recorded using a 16-channel EEG, bipolar EMG, and EOG using the AlphaEEG amplifier and NeuroSpeed software (Alpha Trace Medical Systems, Vienna, Austria).

Sleep scoring: Our PSG was analyzed for sleep stages using the computer-assisted sleep classification system Somnolyer 24 × 7 as developed by the SIESTA group (The SIESTA Group Schlafanalyse GmbH., Vienna, Austria) [9,10] and followed the revised standard criteria described by the American Association for Sleep Medicine (AASM) [11]. The derived sleep features and sleep stages serve as gold standards for the rest of the analyses. Sleep staging for the SC application was realized via a simple image processing of the figures generated by the application; basically, we discretized the SC illustrations into 3 sleep–wake states as suggested by the application in wake, light sleep, and deep sleep (cf. Appendix A for more details).

Statistical analysis: The following five sleep parameters were evaluated: (i) SOL, (ii) SE, (iii) WASO, (iv) TST, and (v) TiB. Importantly, measurements from all the devices were accurately synchronized to the start of the PSG recording. Correlations were computed nonparametrically using Spearman correlations. 

Bland–Altman plots were used to quantify the agreement between the PSG gold standard and the three consumer devices. The measured bias is defined as the mean of the difference between the two-paired measurements. That is, the further this value is from zero, i.e., the line of equality (difference = 0), the higher the error in the measurement. Spearman correlations are used to illustrate systematic linear biases of the devices and are reported at *p* < 0.01 (corrected for the 5 dependent sleep variables analyzed).

Epoch-wise comparison of sleep stages: For analyzing the epoch-by-epoch agreement of the gold standard with the three consumer devices, we always synchronized the recording start with the start of the PSG recording. In case one device started recording after the other (for example, PSG after SC or vice versa), we simply discarded the earlier epochs and started the analysis from the first epoch which was scored by both. As mentioned above, we used the graphs provided by the MB device and the SC application in order to divide 30-s epochs into awake, light sleep, and deep sleep. For the PSG gold standard, light sleep was defined as stages N1 and N2 while deep sleep was defined as the N3 stage. Importantly, when scoring sleep into 3 categories (wake/light sleep/deep sleep) we excluded PSG epochs which were scored as Rapid Eye movement (REM) according to the AASM from the analysis as all 3 devices and applications provide no information about REM (or “dreaming”) sleep. However, in the case of scoring sleep into 2 categories only (wake/sleep), REM epochs were included. We report two main parameters for the epoch-wise agreement: sensitivity and positive predictive value (PPV). Sensitivity (in %) estimates the epoch-by-epoch agreement between MB and SC with the gold standard by measuring the percent of correct classifications (according to the PSG standard) per sleep stage (that is, for example, labelling 79% of all light-sleep detections by the PSG as “light sleep”). The positive predictive value (PPV), on the other hand, is the probability that the assigned state (by the device or application) is indeed that specific state in the gold standard (that is, for example, only 41% of assigned “light sleep” epochs are actually light-sleep epochs and no other sleep states). Cohen’s Kappa (K) as well as the Prevalence adjusted Bias adjusted Kappa (PABAK) were used to assess the pairwise agreement between the devices. A Kappa score < 0.2 is considered a poor agreement while scores between 0.21–0.40 are often considered fair, 0.41–0.60 is considered moderate, and 0.61–0.80 is considered a substantial agreement according to Landis and Koch (1977) [12]. Epoch-wise analysis was computed using IBM SPSS Statistics software (IBM Corp. Released 2017. IBM SPSS Statistics for Windows, Version 25.0. Armonk, NY: IBM Corp.).

## 3. Results

The mean values of the key features of sleep across all participants according to the PSG gold standard were 434.58 ± 95.83 minutes for TiB, 370.12 ± 104.43 minutes for TST, 84.08 ± 13.22% for SE, 25.98 ± 19.35 minutes for SOL, and 39.08 ± 38.43 minutes for WASO. We found no significant difference between laboratory-PSG and home-PSG sleep parameters, but we found a trend towards a lower SOL in home-PSG parameters (Appendix A), which is expected due to the first-night effect [6]. As a first analysis, we simply checked whether the mean sleep values per participant and night correlate between the gold standard and the devices. For TiB, we found good correlation, that is, significant positive associations of the gold standard values with the 3 consumer devices (Appendix A; MB: r = 0.72, *p* = 0.0002; SC: r = 0.67, *p* = 0.02; and MW: r = 0.77, *p* = 0.03). For TST, we only found one moderately positive association for the MB device (r = 0.49, *p* = 0.02), while MW was the only device that showed a significant positive correlation for WASO time (r = 0.78, *p* = 0.02) (see Appendix A). This low correlation is already surprising given that these are simple associations of the mean values per subject, e.g., whether people who take longer to fall asleep in the case of SOL measurements according to the PSG gold standard, also tend to fall asleep later according to the output of one of the consumer devices.

### 3.1. Bland–Altman Plots

We used the Bland and Altman analysis to visualize the degree of agreement between the PSG gold standard and each of the 3 aforementioned sleep trackers (cf. Figure 1, Figure 2 and Figure 3). The most global key features of sleep, namely TIB and TST, are depicted in Figure 1. Looking at the mean difference, there is only a slight bias towards over- or underestimating TIB (MB: 11.17 ± 59.96 min, MW: 11.96 ± 38.48 min, and SC: −5.17 ± 57.57 min). However, the 95% confidence interval also indicates that, for single cases, the devices may still over- or under-estimate TIB by an hour or more (cf. Figure 1A). Mean TST is systematically overestimated by MB by more than an hour (Bias: 69.64 ± 67.43 min) and underestimated by the SC application by a little less than hours on average (Bias: −103.67 ± 85.87 min) (Figure 1B). All 3 tested devices/applications showed inaccurate estimations of SE, with MW giving the best results and showing no systematic over- or underestimation of SE (Figure 2). The mean differences indicate that MB systematically overestimated SE (13.25%) whereas the SC application systematically underestimated SE (26.42%). Interestingly, Spearman correlations indicated that the MB shows greater errors, when the SE values reported by PSG are worse than average. That is, MB has a strong bias towards quantifying SE better than it is. Moreover, we observed a systematic error in the estimation of the WASO time by the MB and the SC but not the MW (Figure 3A). While the MB device underestimates WASO (−33.57 ± 42.84 min), the SC overestimates WASO systematically (89.92 ± 49.90 min). In addition, there is a linear trend in the data showing that MB underestimates WASO time more the longer the actual WASO time gets. The mean difference of the 3 devices/applications to the gold standard is closer to zero for SOL; however, it is to be noted that, here also, the range of possible values is much more limited (36.18 ± 38.37 min for the gold standard). Only MB shows a linear trend with a stronger underestimation of SOL, the longer SOL actually was (as measured by the gold standard) (Figure 3B).

### 3.2. Epoch-Wise Agreement Per Sleep Stage

Table 1 shows the overall and stage-wise agreement between the 30-s epochs scored by our PSG gold standard and by both the MB device and the SC application. Note that MW is disregarded in this respect as standard MW outputs do not provide (or claim to allow) sleep-staging classifications. The overall agreement over all epochs from all subjects (16,350 epochs for MB; 11,243 epochs for MW; and 9504 epochs for SC) between gold-standard PSG scoring and MB was relatively low (53.31%, k = 0.14, PABAK = 0.06) and even lower for the SC device (46.34%, k = 0.18, PABAK = −0.07). Table 1 also illustrates that the highest level of agreement for MB was in determining light sleep (sensitivity = 70.6% and PPV = 57.8%) and that the lowest sensitivity for MB was for detecting wakefulness (sensitivity = 5.5%; PPV= 62.8%). Conversely, SC had moderate sensitivity in identifying awake epochs (sensitivity = 55.6%) and a low PPV value of 24.3%, meaning that only 24.3% of wake-classified epochs are indeed woken states according to the PSG gold standard. On the other hand, SC had low sensitivity in detecting light sleep (40.9%), yet when it classified light sleep, this was the true state in 61.2% of the cases (i.e., PPV = 61.2%). Moreover, for “deep sleep” classification, we found very poor performance for MB (sensitivity = 47.2%, PPV = 43.6%) and poor performance for the SC app (sensitivity = 52.0%, PPV= 53.0%). 

Given this poor performance in correctly classifying sleep stages, we then investigated the ability of these devices and the SC application to simply differentiate between sleep (light and deep sleep) and wakefulness. We also included the scientific MW device (of which the software anyway only provides wake and sleep categories). We then found good overall agreement (OA) for MB and MW (>80%, cf. Table 2) and rather poor OA for the SC app (65.9%). Kappa pairwise agreement indicates a “fair” agreement for MW but poor agreements for the MB and the SC. Specifically, the output shows that the MB and MW devices on the arm and wrist are very good when only “sleep” detection is needed (MB: sensitivity = 99.5%, PPV = 86.8%; MW: sensitivity = 92.9%, PPV = 88.2%). The SC application is as good as the wristband devices in assigning “sleep” to an epoch, as the application is correct in 91.3% of these cases, however, it still misses a third of all sleep epochs (sensitivity = 67.4%). Severe difficulties remain in assigning “awake” epochs by these devices/application and therefore, a proper estimation of overall sleep efficiency or sleep quality remains a challenge. 

Importantly, when we pooled all sleep stages in one stage, i.e. “sleep”, the OA and the PABAK of the MB and the SC increased while their Cohen’s K scores dropped which m indicate a serious bias in the scoring algorithms of the MB device and the SC app. Moreover, when we excluded REM epochs from this analysis, no significant difference in the agreement scores was observed (see Appendix A).

## 4. Discussion

In the present study, we evaluated 2 readily used consumer devices (Mi Band and MotionWatch 8) and one application (Sleep Cycle) for their ability to track sleep. The reason for our selection of such sleep trackers is mainly driven by their dissemination among the public as well as their low cost. We compared these consumer devices to our PSG gold standard which was simultaneously recorded. Overall, we revealed that these devices have an alarmingly low accuracy in scoring sleep in three categories (wake, light sleep, and deep sleep) with the overall agreement ranging between 46.34% for the SC application and 53.02% for the wrist-worn MB. When we tested for the correct classification in only two categories, that is, wake and sleep, the devices of course performed better with an overall agreement of 65.90% for SC, 84.69% for MB, and 81.33% for MW. 

We also showed that all devices and applications had high accuracy in estimating the most global sleep parameter, TiB. Therefore, these devices, in their current status, might be helpful as an objective measure of the time spent in bed at home, preferably in combination with subjective measurements such as sleep diaries. Especially in the case of MW, we need to note that we adjusted the start and the end of the recordings to the PSG gold standard which might overestimate the fidelity of the MW device in measuring TiB. Nevertheless, our MW results are consistent with those reported in previous literature [13,14].

Only for TiB, correlational analysis showed significant positive correlations between the gold standard and all 3 sleep trackers. Although these correlations (see Appendix A) are not sufficient for commenting on the agreement between the sleep trackers and the PSG, they are important to show that even this simple relation does not hold statistically and with alarming disagreements. This raises the question of whether the faulty estimation of values such as TST, SE, WASO, or SOL are due to a priori knowledge of these sleep trackers on the amount of time the average person actually sleeps or needs to fall asleep. If such information is included in the algorithms and outputs of consumer devices, this would explain why the largest errors occur primarily for “non-average” sleep profiles and nights. In line with this observation, previous studies have highlighted the poor performance of sleep trackers when sleep deviates from the average person’s sleep [6,8]. However, to date, this argument remains speculative as the MB and the SC do not allow access to their raw data and are black boxes when it comes to their staging algorithms. Similarly, when comparing the agreement between MB and SC with the PSG gold standard for 3 categories (light sleep, deep sleep, and wake) as compared to 2 categories (sleep vs. wake), we found the expected increase in OA but a drop in the Kappa scores. Especially for MB, looking at the sensitivity scores, we observed extremely low sensitivities in detecting wakefulness (5.5%) and a very high sensitivity in detecting sleep (99.5%). That is, by assigning “sleep” to basically every epoch, the device also cannot miss sleep epochs, yet it of course strongly overestimates sleep and has a vast amount of false alarms for the stage “sleep”. Although MB was the least sensitive between all 3 sleep trackers, it had the highest precision in scoring wakefulness (PPV: 62.8% for MB, 47.8% for MW, and 24.3% for SC). That means that MB does not score awakenings from sleep unless they are almost unmistakable. However, since Cohen’s Kappa is affected by the imbalanced marginal totals in a table [15], masking high levels of agreement, we also reported PABAK which has been shown to be more accurate in such cases [16]. When sleep was scored into 3 categories (wake/light sleep/deep sleep), both K and PABAK were very low (for MB, k = 0.14, PABAK = 0.06 and, for SC, k = 0.18, PABAK = −0.07), confirming the poor agreement between MB/SC and the PSG gold standard. Interestingly, when sleep was scored into two categories (wake/sleep), K scores dropped to half (MB: k = 0.08; SC: k = 0.13) but PABAK increased greatly (MB: PABAK = 0.72; SC: PABAK = 0.30). This might indicate a bias, especially in the MB sleep-scoring algorithm, which again raises the concern whether such a biased output can be of any benefit to the user. It might also be the case that the devices’ bad performance in tracking sleep is due to their inability to capture the subtle changes in sleep architecture throughout the night (see Appendix A).

Regarding the other key parameters, our results raise serious doubts whether such consumer devices and applications can, to date, provide any reliable information about sleep-related health issues. Especially the revealed misjudgment in estimating key features of sleep such as SE, SOL, and WASO are worrisome as they are important diagnostic criteria for quantifying clinically relevant bad sleep and sleep disorders such as insomnia [17]. On the contrary, by providing such inaccurate information, these consumer devices might even risk contributing to worse sleep and life quality as the users may be concerned by the sometimes-negative outputs highlighting bad nights of sleep [5]. 

Our results suggest that wrist-worn devices (MB and MW) tend to have better a performance than mobile-phone applications (SC) in measuring the key features of sleep. This might be attributed to the fact that these devices have direct contact with the body making them more accurate in capturing changes in physiological activity accompanying sleep, and more resilient to environmental factors such as noise or movement from the bed partner, child, pet...etc. In that sense, the combination of both laboratory and at-home PSG recordings in our study is beneficial as this variability reveals whether these sleep trackers can actually detect a range of good to bad sleep patterns independent of the actual sleep environment. This is especially true since the data subset on which the PSG sleep scoring is performed, i.e., the classical AASM channel-set, is identical for both ambulatory and in-lab recordings. Moreover, the semi-automated sleep scoring we used in this study would eliminate any bias when scoring at-home vs. laboratory sleep as it has been shown to have higher agreement with manual scorers than manual scorers among themselves [10]. One very important drawback of these sleep trackers is their inability to provide any information about REM or “dreaming” sleep. Due to an inherent absence of the needed measurements for quantifying REM sleep (most importantly eye movements via EOG and brain activity via EEG), the devices cannot provide the full spectrum of sleep even if the algorithms and sensors would be considerably improved. The incorporation of additional sensors such as an eye or brain electrode might add substantially to the ability of these devices to track and score sleep more accurately and on the long-run, similar to a professional polysomnography in the sleep laboratory. 

An inherent limitation of our evaluation study is that most of our analyses needed to build upon the simple (graphical) outputs of the devices in the form of plots provided for the end user. Moreover, for the tested MB device and SC application, we lost some of the recorded data due to technical reasons unknown to us which lead to differences in the sample size between the three sleep trackers. Additionally, we were not able to directly report the raw data (e.g., heart rate, activity/movements, or sounds) on which these devices and applications build their sleep outputs. Therefore, how these devices and applications generate their results on an epoch-by-epoch basis over the whole night and how they translate their data into sleep stages is unknown to us. We therefore needed to come up with a way to quantify the data and to extract information that can be analyzed statistically (for details, see the Material and Methods section as well as the Appendix A). However, MW allows access to raw data but it is a scientific device that is out of the price range of the usual consumer. Interestingly, this device is likely the most accurate device tested and yet its software only provides two outputs, sleep and wake, as it does not claim to be able to classify sleep in a more fine-grained manner (as compared to MB and SC). However, the main advantage of the commercially available sleep trackers such as MB and SC in their current status is their unmatched affinity with the public, encouraging them to participate in research with a huge impact on the field of sleep research and sleep medicine. These sleep trackers can be valuable for collecting huge amounts of data that otherwise would require a lot of time and money to collect. They can also help us better understand sleep and tackle sleep disorders, given that they provide access to their raw data and analysis algorithms in order to undergo the necessary validation steps [18,19].

In summary, the currently available consumer devices for sleep tracking do not provide reliable information about one’s sleep. However, devices of that kind could be very promising tools for tracking sleep outside the laboratory in the future given that they adhere more to the scientific standards of sleep staging and analysis. Moreover, by refining their algorithms or even by adding more sensors, these devices might be able to reliably monitor and classify sleep across its full range from wakefulness to light sleep, deep sleep, and “REM” dreaming sleep.

## Figures and Tables

**Figure 1 sensors-19-04160-f001:**
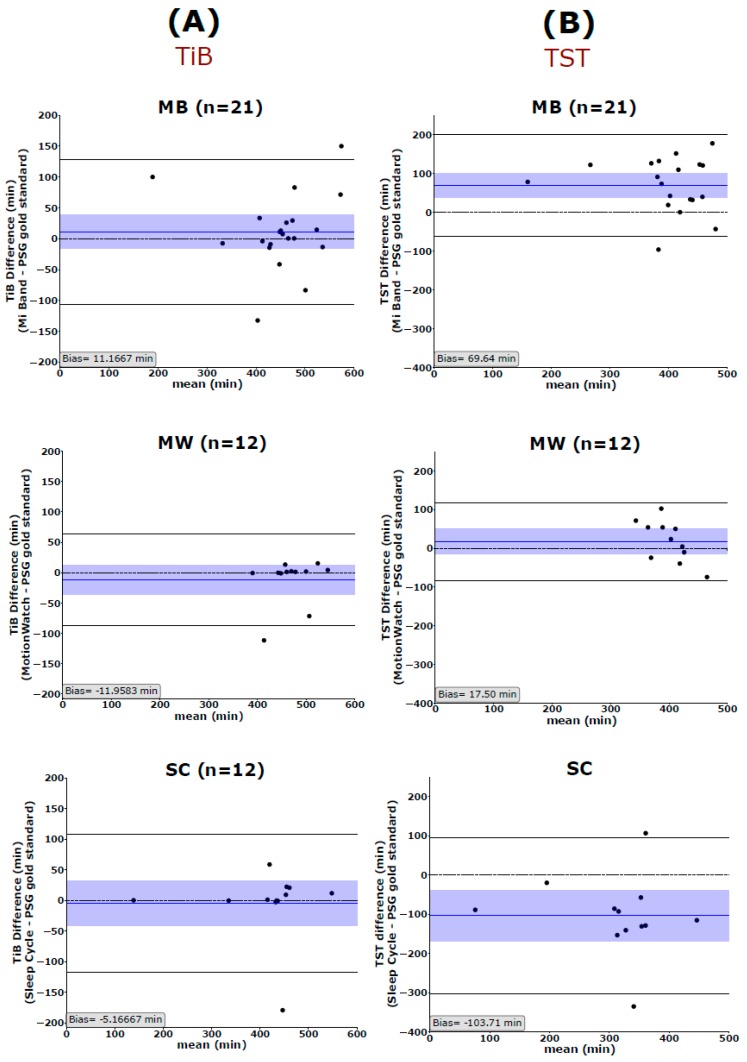
Bland–Altman plots show the agreement of Mi Band (MB), MotionWatch (MW), and Sleep Cycle (SC) with the polysomnography (PSG) in measuring (**A**) time in bed (TiB) but not (**B**) total sleep time (TST). The blue horizontal line represents the mean difference between the two measurements, and the shaded blue area represents the 95% CI of the mean difference. Black horizontal lines mark the 1.96 SD from the mean. The black dashed line is the line of equality (difference = 0). TiB: time in bed, TST: total sleep time, MB: Mi Band, MW: MotionWatch, and SC: Sleep Cycle application.

**Figure 2 sensors-19-04160-f002:**
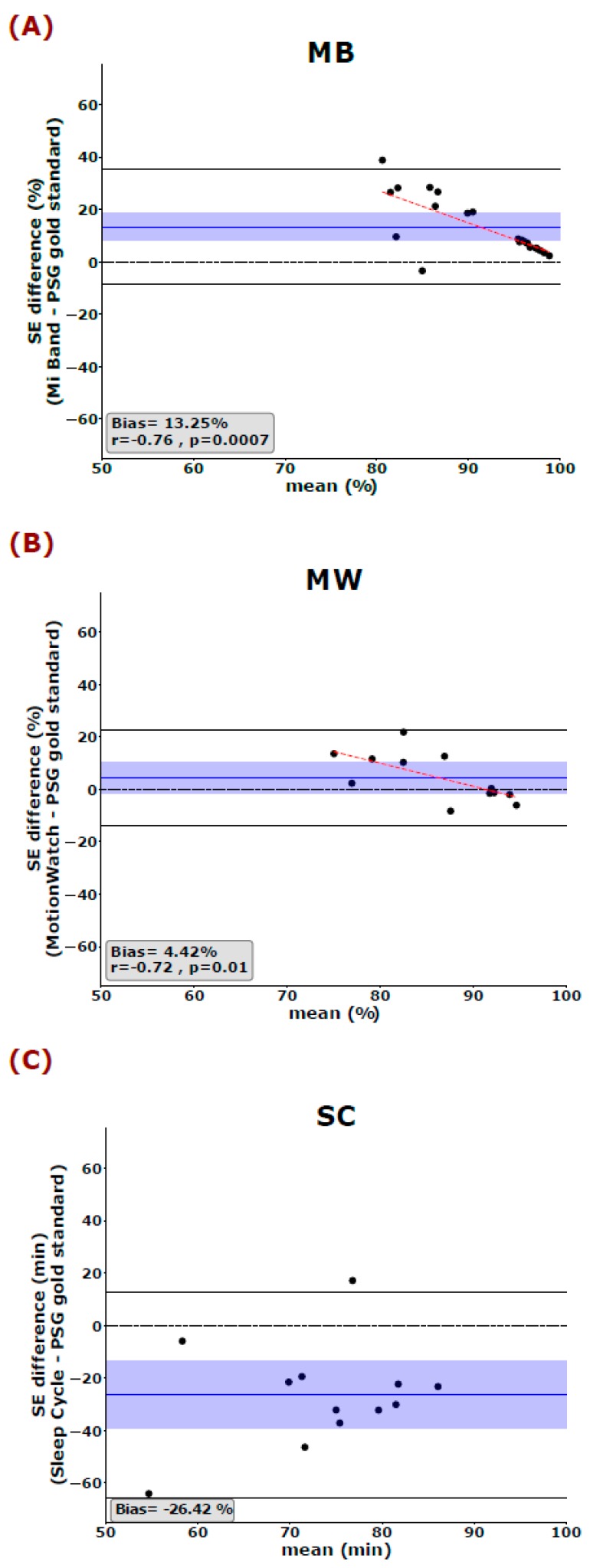
Bland–Altman plots show the PSG gold standard with (**A**) MB, (**B**) MW, and (**C**) SC in measuring sleep efficiency (SE). The blue horizontal line represents the mean difference between the two measurements, and the shaded blue area represents the 95% CI of the mean difference. Black horizontal lines mark the 1.96 SD from the mean. The black dashed line is the line of equality (difference = 0), and the red dashed line represents the Spearman correlation between the difference and the average of the two measurements. SE: Sleep Efficiency, MB: Mi Band, MW: MotionWatch, and SC: Sleep Cycle.

**Figure 3 sensors-19-04160-f003:**
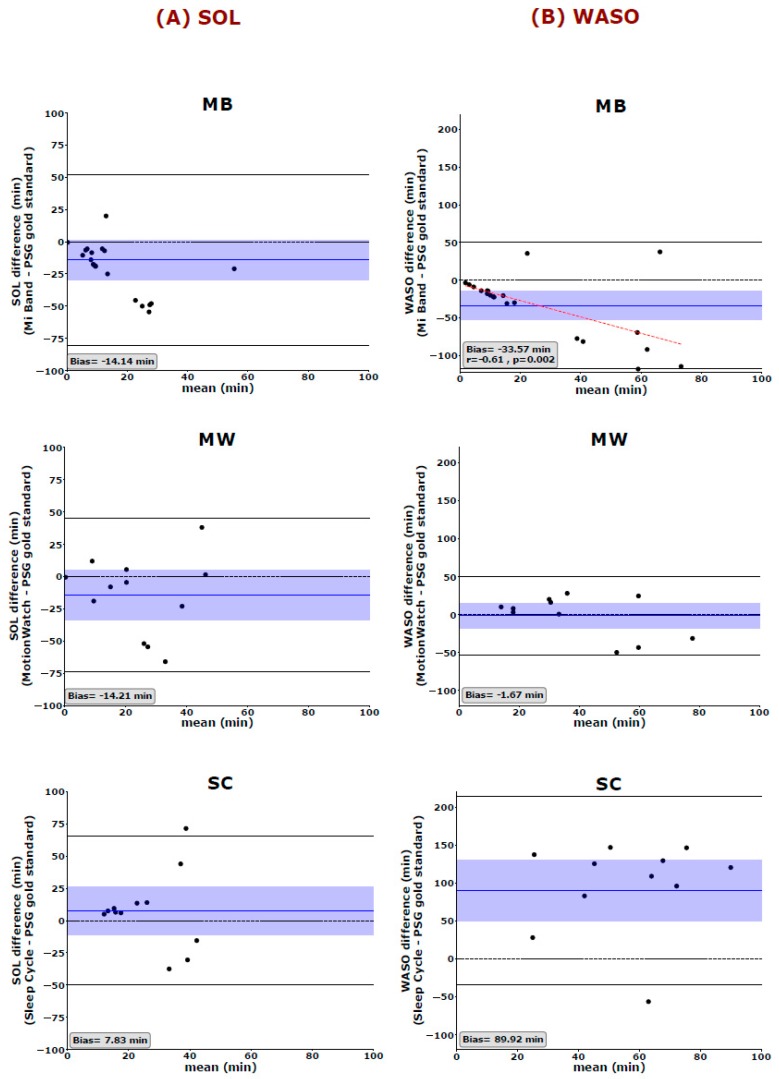
Bland–Altman plots of (**A**) the sleep onset latency (SOL) and (**B**) wake after sleep onset (WASO) measurements show differences between the PSG gold standard and MB, MW, and SC. The blue horizontal line represents the mean difference between the two measurements, and the shaded blue area represents the 95% CI of the mean difference. Black horizontal lines mark the 1.96 SD from the mean. The black dashed line is the line of equality (difference = 0), and the red dashed line represents the Spearman correlation between the difference and the average of the two measurements. SOL: Sleep onset latency, WASO: wake after sleep onset. MB: Mi Band, MW: MotionWatch and SC: Sleep Cycle application.

**Table 1 sensors-19-04160-t001:** Percentages of agreement for 3 stages of sleep scoring (awake/light sleep/deep sleep) between the gold standard (PSG) and the scoring of MB and SC.

	PSG Gold Standard		
	WAKE	LIGHT SLEEP	DEEP SLEEP
Mi Band (MB) staging			
Wake			
% Sensitivity	5.5	0.1	1.5
% PPV	62.8	4.7	32.6
Light sleep			
% Sensitivity	79.2	70.6	51.3
% PPV	18.9	57.8	23.2
Deep sleep			
% Sensitivity	15.3	29.3	47.2
% PPV	7.5	48.9	43.6
Sleep Cycle (SC) staging			
Wake			
% Sensitivity	55.6	37.0	16.9
% PPV	24.3	61.1	14.7
Light sleep			
% Sensitivity	36.4	40.9	31.1
% PPV	14.4	61.2	24.4
Deep sleep			
% Sensitivity	8.0	22.1	52.0
% PPV	4.1	42.8	53.0
Devices/applications	OA (%)	K/PABAK	
Mi Band			
MB	53.31	0.14/0.06	
Sleep Cycle			
SC	46.34	0.18/−0.07	

The agreement is demonstrated by the means of sensitivity (%) as well as by the positive predictive value (PPV). The percentage of overall agreement (OA; %), Cohen’s Kappa coefficient (K) and Prevalence-adjusted Bias-adjusted Kappa (PABAK) are reported for each device.

**Table 2 sensors-19-04160-t002:** Percentages of agreement for 3 stages of sleep scoring (awake/asleep) between the gold standard (PSG) and the scoring of MB and SC.

	PSG Gold Standard	
	WAKE	SLEEP
Mi Band (MB) staging		
Wake		
% Sensitivity	5.5	0.5
% PPV	62.8	37.2
Sleep		
% Sensitivity	94.5	99.5
% PPV	13.2	86.8
Sleep Cycle (SC) staging		
Wake		
% Sensitivity	55.6	32.6
% PPV	19.9	80.1
Sleep		
% Sensitivity	44.4	67.4
% PPV	8.7	91.3
MotionWatch (MW) staging		
Wake		
% Sensitivity	37.5	7.8
% PPV	47.8	52.2
Sleep		
% Sensitivity	62.5	92.9
% PPV	11.5	88.5
Devices/applications	OA (%)	K/PABAK
Mi Band		
MB	86.54	0.08/0.72
Sleep Cycle		
SC	65.90	0.13/0.30
MotionWatch		
MW	83.42	0.33/0.66

The agreement is demonstrated by the means of sensitivity (%) as well as by the positive predictive value (PPV). The percentage of overall agreement (OA; %), Cohen’s Kappa coefficient (K) and Prevalence-adjusted Bias-adjusted Kappa (PABAK) are reported for each device.

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
