# Peer review of "About the Accuracy and Problems of Consumer Devices in the Assessment of Sleep"

_sensors, 2019, doi:10.3390/s19194160_

Round 1

Reviewer 1 Report

Comments:

- There are a number of sleep monitoring devices that are currently available. However, the selection of such device is limited to Mi band, Motionwatch, and Sleep cycle. Are any particular reasons behind the selection of sleep monitoring devices? I suppose it would increase interests of readership.

Authors should review the following articles. Especially, Lee, Jung-Min, et al.’s paper shows comparison results using many commercial devices with many subjects. Thus, authors should cite the following articles and should describe the their results. 

[1] Lee, Jung-Min, et al. "Comparison of Wearable Trackers’ Ability to Estimate Sleep." International journal of environmental research and public health 15.6 (2018): 1265.

https://www.ncbi.nlm.nih.gov/pmc/articles/PMC6025478/

[2] Bhat, Sushanth, et al. "Is there a clinical role for smartphone sleep apps? Comparison of sleep cycle detection by a smartphone application to polysomnography." Journal of Clinical Sleep Medicine 11.07 (2015): 709-715.

http://jcsm.aasm.org/ViewAbstract.aspx?pid=30089

[3] Shelgikar, Anita Valanju, Patricia F. Anderson, and Marc R. Stephens. "Sleep tracking, wearable technology, and opportunities for research and clinical care." Chest 150.3 (2016): 732-743.

This paper assessed two commercial devices and one application to present comparison results for sleep monitoring. There are many commercial sleep monitoring devices. The authors should explain why two devices and one application. Also Other research described heart rate, respiratory rate, etc. But this paper shows sleep time, sleep stages, SE only. The author should explain this reason. 

- The equipment that was used to collect data at home is different from that was used at a lab, which should not be treated as the same type of data without testing it first in terms of data quality or similarity. Please address this issue before analyzing the data further.

- The authors described that sleep parameters (SOL, SE, WASO, TST, TiB) were manually calculated. It seems necessary to provide more detailed criterion about how they were derived manually from data collected by the devices and the app rather than describing it merely “manually calculated”.

- Again, the authors described that they divided two-state output (sleep/waking) into three categories (awake, light sleep, and deep sleep), but information on how to achieve that was not given. It seems that the equations or criterion on how to do so are required to ensure reproducibility of the outcomes.

- The data collected for each device and an application are not balanced. Statistical influences of such imbalance in the samples must be addressed in order to provide readers sufficient information to objectively interpret the experimental results.

Author Response

First of all, we would like to thank all reviewers for their time and effort as well as their interest in our work. The review process has substantially improved our manuscript and we hope that our responses are to the full satisfaction of the reviewers.

Reviewer 1

There are a number of sleep monitoring devices that are currently available. However, the selection of such device is limited to Mi band, Motionwatch, and Sleep cycle. Are any particular reasons behind the selection of sleep monitoring devices? I suppose it would increase interests of readership. We thank Reviewer 1 for that remark and would like to mention that we have added a few lines in the original manuscript describing the motivation behind our choice of the tested devices and applications (Introduction: Line 79, Discussion: Line ). However, we would also like to respond directly and thoroughly to the reviewer here:

In this paper we validated one commercial activity tracker; The MI band (Xiaomi, Beijing, China), one research-grade accelerometer; The MotionWatch (CamNTech, Cambridge, UK) and one mobile-phone application; Sleep Cycle (Northcube, Gutenberg, Sweden). This choice was made and we were aiming at assessing different kinds of sleep trackers available at various levels. That is we assessed a much used (and at sleep congresses much discussed) an freely available phone application (Sleep Cycle), a much used and low cost, commercially available sleep tracker (Xiaomi Mi Fit) as well as a high level, high cost, in sleep lab readily used scientifically used accelerometer. So the popularity of these devices and applications was one of our key criteria. The mi band and the Sleep Cycle App are currently being used by at least millions of users worldwide, as stated on their website, and the MotionWatch is, to our knowledge, one of the most popular accelerometer used in research. We thought that it is quite intriguing how these devices are performing when compared to the current PSG gold-standard especially as there seems to be no attempt to scientifically validate or review these devices by the manufactures themselves. It is also worth mentioning that our laboratory has a history and interest in assessing such devices since years (Griessenberger et al., 2013; Voinescu et al., 2014). Moreover, we need to mention that we are currently testing also other commercial and research-grade sleep trackers that promise to allow at least some minimal access to their “raw data” for validation. These time-consuming tests however are pending and are not expected to be finished before spring next year.

[1] Voinescu, B.I., Wislowska, M., Schabus, M., 2014. Assessment of SOMNOwatch plus EEG for sleep monitoring in healthy individuals. Physiol. Behav. 132,73–78.

[2] Griessenberger, H., Heib, D.P.J., Kunz, A.B., Hoedlmoser, K., Schabus, M., 2013. Assessment of a wireless headband for automatic sleep scoring. Sleep Breath 17, 747–752.

Authors should review the following articles. Especially, Lee, Jung-Min, et al.’s paper shows comparison results using many commercial devices with many subjects. Thus, authors should cite the following articles and should describe their results.

[1] Lee, Jung-Min, et al. "Comparison of Wearable Trackers’ Ability to Estimate Sleep." International journal of environmental research and public health 15.6 (2018): 1265. https://www.ncbi.nlm.nih.gov/pmc/articles/PMC6025478/

[2] Bhat, Sushanth, et al. "Is there a clinical role for smartphone sleep apps? Comparison of sleep cycle detection by a smartphone application to polysomnography." Journal of Clinical Sleep Medicine 11.07 (2015): 709-715. http://jcsm.aasm.org/ViewAbstract.aspx?pid=30089

[3] Shelgikar, Anita Valanju, Patricia F. Anderson, and Marc R. Stephens. "Sleep tracking, wearable technology, and opportunities for research and clinical care." Chest 150.3 (2016): 732-743.

We thank the reviewer for drawing our attention to these publications. We have incorporated the findings of these articles in our manuscript and kindly refer to references [6,7] in the introduction and reference [19] in the discussion.

This paper assessed two commercial devices and one application to present comparison results for sleep monitoring. There are many commercial sleep monitoring devices. The authors should explain why two devices and one application. Also other research described heart rate, respiratory rate, etc. But this paper shows sleep time, sleep stages, SE only. The author should explain this reason. Please refer to point 1 above. We also now mention in the introduction and the discussion the reasons for our choice. Moreover, we actually do not provide data for heart or respiratory rate as we cannot derive time lines (like 30sec by 30sec timelines of such data) of these “black box” devices and apps; furthermore our main interest in this study is the claimed output (sleep staging, and sleep parameters) of these devices rather than the raw data on which they may build upon (Note that it is actually also quite unknown how far these devices rely on movement, heart rate or HRV data, or sound).

Introduction (Line 79): “We aimed at choosing some of the relatively cheap and highly popular commercial sleep trackers, the Mi band and the Sleep Cycle application, as well as at least one research-grade sleep tracker, the Motionwatch. By choosing these three sleep trackers we are able to test at least one device/application from each class of consumer devices available in the market.”

Discussion (Line 345): “The reason for our current selection of such sleep trackers is mainly driven by their dissemination in the public as well their low cost enabling their widespread use in the population.”

The equipment that was used to collect data at home is different from that was used at a lab, which should not be treated as the same type of data without testing it first in terms of data quality or similarity. Please address this issue before analyzing the data further. We thank reviewer 1 for this important remark. Our response to this comment is twofold:

By drawing a simple comparison between the data recorded using hdEEG in the laboratory and the ambulatory EEG system for the home recordings we were not able to find any statistical differences in the sleep parameters as depicted in Table S4 below (also added to the supplements). We do find a trend to shorter sleep onset latency at home as in lab as expected as due to the first night effect in the laboratory. However, if it comes down to the methodological differences between the hdEEG system and the ambulatory EEG one, there shouldn’t be any difference in the signal used for sleep scoring as our laboratory uses a fixed standard procedure for sleep scoring to ensure the quality and the reproducibility of the data and the analyses. That is, we always used the same set of electrodes for sleep staging no matter which device we are using ambulatory or in lab (that is, electrodes C3_A2, C4_A1, F3_A2, F4_A1, O1_A2, O2_A1, EOG and bipolar EMG). This setup is the recommended setup by the American Association for Sleep Medicine (AASM & Iber 2007) to ensure replicability across labs internationally. Please also note that for the agreement of our PSG gold standard with the consumer devices it should not matter at all whether we record in-lab sleep or ambulatory sleep at home. Actually this setup even helps us to generalize that no matter if we record better sleep (at home) or worse sleep in lab the consumer devices are reliably (or not reliably) able to monitor sleep. This variance is actually of a huge benefit as well to the current analysis as we see clearly that the devices “expect” a certain sleep quality independent of the underlying data and therefore are not valid for detecting sleep especially if people have a night outside of the expected mean values.

Reference:

American Academy of Sleep Medicine. (2007). The AASM manual for the scoring of sleep and associated events: rules, terminology and technical specifications. Westchester, IL: American Academy of Sleep Medicine, 23.

Table S4. A comparison between sleep parameters as measured by the laboratory hdEEG and the at-home ambulatory EEG systems.

Parameter

In-lab hdEEG (n=17)

Ambulatory EEG (n=8)

P*

TiB

452.29 ± 81.78  min

396.94 ± 117.54 min

0.26

TST

378.5 ± 95.21 min

352.31 ± 127 min

0.61

SOL

30 ± 20.49 min

17.44 ± 14.19 min

0.09+

WASO

44.44 ± 38.66 min

27.69 ± 37.80 min

0.32

SE

82.41 ± 13.83%

87.63 ± 11.89%

0.35

*Independent sample t-test without assuming equal variances (Welch approximation t-test). + p<0.1 or a statistical trend. TiB: time in bed; TST: total sleep time; SOL: sleep onset latency; WASO: wake after sleep onset; SE: sleep efficiency.

The authors described that sleep parameters (SOL, SE, WASO, TST, TiB) were manually calculated. It seems necessary to provide more detailed criterion about how they were derived manually from data collected by the devices and the app rather than describing it merely “manually calculated”. We thank the reviewer for this observation. We would like to clarify that some of the parameters were extracted from the devices and/or applications whenever possible, however, some values which were not provided by these devices were manually calculated as outlined in the method section with the subtitle “instruments” on page 3, line 100. We have addressed this point in the main article as following: For the Mi band: “…For our analysis, we used the hypnogram (graph) produced by the Mi Fit software to extract the exact timings of the start and the end of sleep, the start and the end of each sleep stage (light sleep/deep sleep) as well as the duration of the wake WASO in minutes. Using these data we calculated the TiB as the difference between the start and the end times reported in the software and the TST as the total amount of time spent asleep (both light or deep sleep) during this period. SOL is defined as the difference between the time when the MB start scoring sleep and the time of the 1st N1 epoch as scored by the PSG gold standard. Finally, SE was calculated as follows; SE= (TST/TiB)*100. The 30s epochs for the epoch-by-epoch agreement were extracted manually by fragmenting the segments of wake/light sleep/Deep sleep in the graph into 30s epochs.” For the MotionWatch: “…TiB, TST, SOL and SE are calculated automatically by the software and WASO was calculated manually. TiB is defined as the the total elapsed time between the “Lights Out” and “Got Up” times. TST is defined as the total elapsed time between the “Fell Asleep” and “Woke Up” times. SOL is defined as the time between “Lights Out” and “Fell Asleep” and SE as the “Actual Sleep Time” divided by “Time in Bed” in percentage. Finally, WASO was determined by calculating the amount of epochs scored as awake after the first epoch of sleep. 30s epoch data provided by the software was used to calculate the epoch-by-epoch agreement with gold standard.” For Sleep Cycle: “…As even the premium version of the application provides no access to the raw data we performed image analysis of the hypnogram provided by the application (see suppl. material and supplementary figure S1 for more information) as a workaround to this problem. Through this workaround we were able to extract 30s-epoch information on sleep sleep scoring (Wake/Light sleep/Deep sleep) which was also used for measuring sleep parameters. TiB was calculated based on the “went to sleep” and woke up times reported by the application. TST was calculated by subtracting all the “awake” epochs from the TiB. SOL was calculated from by summing up all the awake 30s epochs before the first light sleep epoch. WASO was calculated by summing up all the awake epochs that lie between the first light sleep epoch and the “woke up” time. SE = (TST/TiB)*100.”

Again, the authors described that they divided two-state output (sleep/waking) into three categories (awake, light sleep, and deep sleep), but information on how to achieve that was not given. It seems that the equations or criterion on how to do so are required to ensure reproducibility of the outcomes.

We thank the reviewer for this remark and we would like to clarify that we have included a more detailed step-wise description of how we extracted and analyzed the data from the 3 trackers. This information has been added to the main manuscript in the material and methods section under the subtitle “Instruments” line 100. We also added this section here for the convenience of the reviewer:

“Instruments: Mi-Band (MB; Xiaomi, Beijing, China): The Mi Band is a wrist-worn commercial activity tracker (15.7mm X10.5mmX 40.3mm, and weighs 7.0g) that detects sleep using a combination of two sensors; i) a proximity sensor and ii) actigraphy (actimetric sensor). While the proximity sensor detects contact with the skin, the actimetric sensor detects body movements to count sleep time and differentiate between light and deep. These data are then converted to a hypnogram using the Mi Fit software (algorithm v1.1.14,   Anhui Humai Infomration Techniology CO., Ltd., Hefei, China). We used the Mi band 2, however, some of the recordings were done using the Mi band 3 which, as advised by Xiaomi developers on their website, has no hardware update from the Mi band 2 that might influence the results. For our analysis, we used the hypnogram (graph) produced by the Mi Fit software to extract the exact timings of the start and the end of sleep, the start and the end of each sleep stage (light sleep/deep sleep) as well as the duration of the wake WASO in minutes. Using these data we calculated the TiB as the difference between the start and the end times reported in the software and the TST as the total amount of time spent asleep (both light or deep sleep) during this period. SOL is defined as the difference between the start time of the PSG recording and the time when the MB starts recording sleep “Light sleep”. Finally, SE was calculated as follows; SE= (TST/TiB)*100. The 30s epochs for the epoch-by-epoch agreement were extracted manually by fragmenting the segments of wake/light sleep/Deep sleep in the graph into 30s epochs.

MotionWatch8 (MW; CamNTech, Cambridge, UK):  The MW is a research-grade actigraphy with a built-in ambient light sensor (Dimesnions: 36mm l X 28.2 w X 9.4mm d excluding strap, and weighs 9.1 grams including battery but excluding strap; Sensor: Triaxial, MEMs technology, 0.01g to 8g range, 3 – 11Hz). For differentiating between sleep and wake epochs we used the MotionWare software (v1.1.20, empire Software GmbH, Cologne, Germany) which utilizes an algorithm that depends on thresholding. Briefly, this thresholding algorithm assigns an activity score to each epoch by totalizing the epoch in question and those surrounding it using weighting factors. If the activity score of an epoch is above a predefined threshold, then the epoch is scored as wake, otherwise it is scored as sleep. This activity score is dependent upon the sampling epoch-length. Since we used 30s epochs, the activity scores are calculated as follows: the number of movements in the epoch being scored is multiplied by 2 and that of the epochs within 2 minutes of this epoch (1 minute before and 1 minute after) are multiplied by 0.2. The activity score of this epoch is the sum of these weighted values. The MW has 3 thresholding options; low, medium and high corresponding to 80, 40, 20 activity score respectively. For our analysis we used the high threshold value as it was used for the validation of the MW. Moreover, an additional level of scoring is also done based on the movements detected by the MW per epoch. If there is more than 2 movements in a 30s epoch, this epoch is scored as mobile (awake). The detection of the start of sleep is based on 10-minute segments and is totally independent of the sleep/wake scoring described earlier as the threshold is 3 activity counts in a 30s epoch. The process starts by looking at the first 10 minutes (20 30s-epochs) after lights-out time (which was synchronized to the PSG). Each epoch is tested against the threshold (i.e. 3 counts) and the number of epochs exceeding the threshold is counted. If this number is greater than 2 then the process is repeated 1 minute later. This process continues until a 10-minute block that fulfils the criteria is found marking the start of sleep.  Detecting the end of sleep is done using the same procedure, however, instead of 10-minute segments of maximum 2 epochs containing 3 or more activity counts, 5-minute segments of maximum 2 epochs containing 5 or more activity counts marks the end of sleep. TiB, TST, SOL and SE are calculated automatically by the software and WASO was calculated manually. TiB is defined as the the total elapsed time between the “Lights Out” and “Got Up” times. TST is defined as the total elapsed time between the “Fell Asleep” and “Woke Up” times. SOL is defined as the time between “Lights Out” and “Fell Asleep” and SE as the “Actual Sleep Time” divided by “Time in Bed” in percentage. Finally, WASO was determined by calculating the amount of epochs scored as awake after the first epoch of sleep. 30s epoch data provided by the software was used to calculate the epoch-by-epoch agreement with gold standard.

Sleep Cycle (SC; NorthCube, Gothenberg, Sweden): SC is a mobile phone application that is available on android-based as well as IOS-based devices. SC is a smart alarm clock that tracks your sleep patterns and wakes you up during light sleep. SC tracks sleep throughout the night and use a 30-minute window that ends up with the desired alarm time during which the alarm goes off at the lightest possible sleep stage (i.e. light sleep). SC scores sleep through motion detection via one of two motion detection modes; i) microphone: which uses the built-in microphone is used to analyze movements or ii) accelerometer which uses the built-in accelerometer. SC tracks movements through the night and uses them to detect and score sleep as well as plotting a graph (hypnogram). For our analysis we used the SC on both IOS (v5.7.1) and Android (v3.0.1). We were advised by the developers (Northcube, GOT, Sweden) that there is no difference between the two versions in sleep scoring algorithms). We used the recommended settings for recording, that is, we used the microphone to detect movements and we placed the phone next to the participants with the microphone facing the participant and the charger plugged in. Although we used the premium version, we were not able to find any information on the algorithm by which SC is able to dissociate between sleep and wakefulness as well as to classify of sleep into light and deep sleep. As even the premium version of the application provides no access to the raw data we performed image analysis of the hypnogram provided by the application (see suppl. material and supplementary figure S1 for more information) as a workaround to this problem. Through this workaround we were able to extract 30s-epoch information on sleep sleep scoring (Wake/Light sleep/Deep sleep) which was also used for measuring sleep parameters. TiB was calculated based on the “went to sleep” and woke up times reported by the application. TST was calculated by subtracting all the “awake” epochs from the TiB. SOL was calculated from by summing up all the awake 30s epochs before the first light sleep epoch. WASO was calculated by summing up all the awake epochs that lie between the first light sleep epoch and the “woke up” time. SE = (TST/TiB)*100.”

The data collected for each device and an application are not balanced. Statistical influences of such imbalance in the samples must be addressed in order to provide readers sufficient information to objectively interpret the experimental results. We thank the reviewer for that observation. This observed imbalance in the samples is mainly because we lost some data during the recordings from some applications and devices due to technical reasons also unknown to us. Yet, we now state more clearly that there are some differences in the number of observations between the devices due to technical difficulties and data loss in the various recordings (see discussion Line 431):

“An inherent limitation of our evaluation study is that most of our analysis needed to build upon the simple (graphical) outputs of the devices in form of plots provided for the end-user. For the tested MB device and SC app there is no way to access the raw data we lost some of the recorded data due to technical reasons unknown to us which lead to differences in the sample size for each of the devices. Moreover, we were not able to directly report the raw data (e.g. heart rate, activity/movements or sounds) on which these devices and applications build their sleep outputs. Therefore, how these devices and applications generate their results on an epoch-by-epoch basis over the whole night and how they translate their data into sleep stages remains unknown.”

Reviewer 2 Report

The current paper aimed at assessing the accuracy in detecting sleep-wake as well as sleep stages of one commercially-available sleep tracker (MB), a widely-used actigraph (MW), and one smartphone app (SC) against gold standard polysomnography (PSG). The authors showed that all devices had some degrees of error against PSG. In particular, the sleep tracker and the app showed an unsatisfactory agreement with PSG, as released by Bland-Altman plots and Cohen’s Kappa. The actigraph was the most reliable instrument compared to PSG, showing the standard bias observed in most of the actigraphs (i.e., underestimation of SOL and WASO and overestimation of TST).

The paper is timely and important due to the massive amount of non-validated sleep trackers on the market. I have only some suggestions that in my view my help clarifies some methodological aspects of the paper.

1- Please consider to add more information about the three devices. For example, how do they collect sleep data (e.g., only motion, motion plus pulse velocity), whether they wrist-worn devices, where was the smartphone positioned (e.g., under the pillow).

2- It was not clear to me the sampling frequency of the 3 devices. Specifically, were the stages (either sleep staging or only sleep/wake) extracted in 30s as for the PSG? Of with a different time resolution? In the latter case, how the authors dealt with the different temporal windows for the epoch-by-epoch analysis? Please clarify.

3- It is not clear to me how the MB data were extracted. In a similar way as the SC (extracting the values from the hypnogram on the screen) or using some raw data? Please clarify.

4 - I think there is an error in the sentence “For the MW measurements, the sensitivity threshold was set to 20 activity counts and adjusted the lights off and lights on times according to sleep diaries as usually done for scientific actigraph measurements. SOL, SE, TST, WASO, and TiB values are then used as provided by the MotionWare software” (p.3, l. 102-105). Since the data were collected in the lab, and synchronized with PSG, it is not clear whether the authors used sleep diaries for.

5 - The Authors used Person correlations to assess the relationship between the sleep parameters of the devices and the PSG. However, Person correlation is not the best method for assessing agreement between different measures (see Bland, J. M., & Altman, D. G. (1999). Measuring agreement in method comparison studies. Statistical methods in medical research, 8(2), 135-160). Please consider to use more appropriate analyses, such as intraclass correlations, or to clearly state, for non-expert readers, that Pearson correlations cannot provide any information about the measurement error or of the agreement between devices

6 - Looking at the Bland-Altman plots, it seems to me that both the MB and the SC showed also a problem of heteroscedasticity (i.e., increase error as a function of the magnitude of the measured value) for SOL and WASO. The same problem, but only for SOL, seems to happen for the MW. Please consider to mention and discuss these results.

7 - The authors used Cohen’s Kappa coefficient to assess the agreement between the devices, reporting very low values. Please consider that for sleep, Cohen’s Kappa is a very liberal index, since there is a higher proportion of sleep epochs compared to wake epochs. This can lead to ‘‘the first paradox of kappa statistic’’ (see Feinstein, A. R., & Cicchetti, D. V. (1990). High agreement but low kappa: I. The problems of two paradoxes. Journal of clinical epidemiology43(6), 543-549. and Cicchetti, D. V., & Feinstein, A. R. (1990). High agreement but low kappa: II. Resolving the paradoxes. Journal of clinical epidemiology43(6), 551-558.), that is, when two measures show high agreement but a low kappa. A more appropriate and conservative measure is the Prevalence- And Bias-Adjusted Kappa (PABAK), which gives balanced weight to sleep and wake epochs (see also Cellini, N., Buman, M. P., McDevitt, E. A., Ricker, A. A., & Mednick, S. C. (2013). Direct comparison of two actigraphy devices with polysomnographically recorded naps in healthy young adults. Chronobiology International30(5), 691-698.). Please consider reanalyzing the data with this approach or, as an alternative, to discuss that the results have been obtained with a low conservative approach.  

8- In the discussion, the authors say that “devices and applications had high accuracy in estimating the most global sleep parameter, namely TiB. This makes these devices a helpful tool for objectively measuring the time spent in bed at home rather than relying solely on subjective measures such as daily sleep diaries” (p.12, l.251-3). In my opinion, this sentence may be toned done, since, if I understood correctly, all the devices were turned on just before going to sleep in the same condition (lab) for most of the subjects, reducing the variability of TiB. To claim that TiB detection was accurate the authors may show the TiB detection in the subsample of participants who slept at home. That wm

9- Please consider to discuss the results in lights of the finding of a recent review of all the validated sleep trackers against PSG (de Zambotti, M., Cellini, N., Goldstone, A., Colrain, I. M., & Baker, F. C. (2019). Wearable Sleep Technology in Clinical and Research Settings. Medicine and science in sports and exercise, 51(7):1538-1557)

Author Response

First of all, we would like to thank all reviewers for their time and effort as well as their interest in our work. The review process has substantially improved our manuscript and we hope that our responses are to the full satisfaction of the reviewers.

Reviewer 2

The current paper aimed at assessing the accuracy in detecting sleep-wake as well as sleep stages of one commercially-available sleep tracker (MB), a widely-used actigraph (MW), and one smartphone app (SC) against gold standard polysomnography (PSG). The authors showed that all devices had some degrees of error against PSG. In particular, the sleep tracker and the app showed an unsatisfactory agreement with PSG, as released by Bland-Altman plots and Cohen’s Kappa. The actigraph was the most reliable instrument compared to PSG, showing the standard bias observed in most of the actigraphs (i.e., underestimation of SOL and WASO and overestimation of TST).

The paper is timely and important due to the massive amount of non-validated sleep trackers on the market. I have only some suggestions that in my view my help clarifies some methodological aspects of the paper.

Please consider to add more information about the three devices. For example, how do they collect sleep data (e.g., only motion, motion plus pulse velocity), whether they wrist-worn devices, where was the smartphone positioned (e.g., under the pillow). We thank reviewer 2 for that important remark. We addressed this issue in the methods section and a whole sub-section is added (page 3 Line 100) to describe the 3 devices /application used in more details. It reads as follows: “Instruments: Mi-Band (MB; Xiaomi, Beijing, China): The Mi Band is a wrist-worn commercial activity tracker (15.7mm X10.5mmX 40.3mm, and weighs 7.0g) that detects sleep using a combination of two sensors; i) a proximity sensor and ii) actigraphy (actimetric sensor). While the proximity sensor detects contact with the skin, the actimetric sensor detects body movements to count sleep time and differentiate between light and deep. These data are then converted to a hypnogram using the Mi Fit software (algorithm v1.1.14, Anhui Humai Infomration Techniology CO., Ltd., Hefei, China). We used the Mi band 2, however, some of the recordings were done using the Mi band 3 which, as advised by Xiaomi developers on their website, has no hardware update from the Mi band 2 that might influence the results. For our analysis, we used the hypnogram (graph) produced by the Mi Fit software to extract the exact timings of the start and the end of sleep, the start and the end of each sleep stage (light sleep/deep sleep) as well as the duration of the wake WASO in minutes. Using these data we calculated the TiB as the difference between the start and the end times reported in the software and the TST as the total amount of time spent asleep (both light or deep sleep) during this period. SOL is defined as the difference between the start time of the PSG recording and the time when the MB starts recording sleep “Light sleep”. Finally, SE was calculated as follows; SE= (TST/TiB)*100. The 30s epochs for the epoch-by-epoch agreement were extracted manually by fragmenting the segments of wake/light sleep/Deep sleep in the graph into 30s epochs.

MotionWatch8 (MW; CamNTech, Cambridge, UK):  The MW is a research-grade actigraphy with a built-in ambient light sensor (Dimesnions: 36mm l X 28.2 w X 9.4mm d excluding strap, and weighs 9.1 grams including battery but excluding strap; Sensor: Triaxial, MEMs technology, 0.01g to 8g range, 3 – 11Hz). For differentiating between sleep and wake epochs we used the MotionWare software (v1.1.20, empire Software GmbH, Cologne, Germany) which utilizes an algorithm that depends on thresholding. Briefly, this thresholding algorithm assigns an activity score to each epoch by totalizing the epoch in question and those surrounding it using weighting factors. If the activity score of an epoch is above a predefined threshold, then the epoch is scored as wake, otherwise it is scored as sleep. This activity score is dependent upon the sampling epoch-length. Since we used 30s epochs, the activity scores are calculated as follows: the number of movements in the epoch being scored is multiplied by 2 and that of the epochs within 2 minutes of this epoch (1 minute before and 1 minute after) are multiplied by 0.2. The activity score of this epoch is the sum of these weighted values. The MW has 3 thresholding options; low, medium and high corresponding to 80, 40, 20 activity score respectively. For our analysis we used the high threshold value as it was used for the validation of the MW. Moreover, an additional level of scoring is also done based on the movements detected by the MW per epoch. If there is more than 2 movements in a 30s epoch, this epoch is scored as mobile (awake). The detection of the start of sleep is based on 10-minute segments and is totally independent of the sleep/wake scoring described earlier as the threshold is 3 activity counts in a 30s epoch. The process starts by looking at the first 10 minutes (20 30s-epochs) after lights-out time (which was synchronized to the PSG). Each epoch is tested against the threshold (i.e. 3 counts) and the number of epochs exceeding the threshold is counted. If this number is greater than 2 then the process is repeated 1 minute later. This process continues until a 10-minute block that fulfils the criteria is found marking the start of sleep.  Detecting the end of sleep is done using the same procedure, however, instead of 10-minute segments of maximum 2 epochs containing 3 or more activity counts, 5-minute segments of maximum 2 epochs containing 5 or more activity counts marks the end of sleep. TiB, TST, SOL and SE are calculated automatically by the software and WASO was calculated manually. TiB is defined as the the total elapsed time between the “Lights Out” and “Got Up” times. TST is defined as the total elapsed time between the “Fell Asleep” and “Woke Up” times. SOL is defined as the time between “Lights Out” and “Fell Asleep” and SE as the “Actual Sleep Time” divided by “Time in Bed” in percentage. Finally, WASO was determined by calculating the amount of epochs scored as awake after the first epoch of sleep. 30s epoch data provided by the software was used to calculate the epoch-by-epoch agreement with gold standard.

Sleep Cycle (SC; NorthCube, Gothenberg, Sweden): SC is a mobile phone application that is available on android-based as well as IOS-based devices. SC is a smart alarm clock that tracks your sleep patterns and wakes you up during light sleep. SC tracks sleep throughout the night and use a 30-minute window that ends up with the desired alarm time during which the alarm goes off at the lightest possible sleep stage (i.e. light sleep). SC scores sleep through motion detection via one of two motion detection modes; i) microphone: which uses the built-in microphone is used to analyze movements or ii) accelerometer which uses the built-in accelerometer. SC tracks movements through the night and uses them to detect and score sleep as well as plotting a graph (hypnogram). For our analysis we used the SC on both IOS (v5.7.1) and Android (v3.0.1). We were advised by the developers (Northcube, GOT, Sweden) that there is no difference between the two versions in sleep scoring algorithms). We used the recommended settings for recording, that is, we used the microphone to detect movements and we placed the phone next to the participants with the microphone facing the participant and the charger plugged in. Although we used the premium version, we were not able to find any information on the algorithm by which SC is able to dissociate between sleep and wakefulness as well as to classify of sleep into light and deep sleep. As even the premium version of the application provides no access to the raw data we performed image analysis of the hypnogram provided by the application (see suppl. material and supplementary figure S1 for more information) as a workaround to this problem. Through this workaround we were able to extract 30s-epoch information on sleep sleep scoring (Wake/Light sleep/Deep sleep) which was also used for measuring sleep parameters. TiB was calculated based on the “went to sleep” and woke up times reported by the application. TST was calculated by subtracting all the “awake” epochs from the TiB. SOL was calculated from by summing up all the awake 30s epochs before the first light sleep epoch. WASO was calculated by summing up all the awake epochs that lie between the first light sleep epoch and the “woke up” time. SE = (TST/TiB)*100.”

It was not clear to me the sampling frequency of the 3 devices. Specifically, were the stages (either sleep staging or only sleep/wake) extracted in 30s as for the PSG? Or with a different time resolution? In the latter case, how do the authors dealt with the different temporal windows for the epoch-by-epoch analysis? Please clarify.

We thank reviewer 2 for that important question. For the MotionWatch we had the option to choose the length of epochs for the analysis and so we extracted 30s epoch data. For the Sleep Cycle as we had no access to raw data, the method we developed to analyze the figures (hypnograms) produced by the application eventually also produced 30s epochs (please see supplementary figure S1). For the Mi Band, however, the staging is based on 1 minute epochs, so we had to up-sample the data of the Mi band such that 1 minute of light sleep for example is translated in to two 30s-epochs of light sleep during that minute, etc. We added this information in the “Instrument” subsection of the material and methods.

It is not clear to me how the MB data were extracted. In a similar way as the SC (extracting the values from the hypnogram on the screen) or using some raw data? Please clarify. We were not able to access raw data neither for the Mi band or the sleep cycle. Therefore, we analyzed the final hypnogram produced by the software. For the Mi band the hypnogram specified the time of the start and the end of each stage in minutes (for example: from 11:30 to 11:31 light sleep & from 11:32 to 11:33 deep sleep). We used these timings on the hypnogram provided by the Mi Fit software for calculating sleep parameters as well as for the epoch-by-epoch analysis. For the SC the hypnogram produced by the application gives no timing values but for the “went to bed” and the “woke up” times. Therefore we developed our own method (see suppl. material, figure S1) to extract the 30s epoch information from the hypnograms which was then used for the epoch-by-epoch as well as the sleep parameters analyses. This is also explained better in the materials and methods section page 3 line 100.

I think there is an error in the sentence “For the MW measurements, the sensitivity threshold was set to 20 activity counts and adjusted the lights off and lights on times according to sleep diaries as usually done for scientific actigraph measurements. SOL, SE, TST, WASO, and TiB values are then used as provided by the MotionWare software” (p.3, l. 102-105). Since the data were collected in the lab, and synchronized with PSG, it is not clear whether the authors used sleep diaries for. Thanks to the reviewer for pointing out the ambiguity of the statement we used. We would like to clarify that we indeed synchronized the start of the MotionWatch to the start of the PSG recording. However, as a secondary confirmatory step, and as we had information on the lights on and lights off times reported in the sleep diaries of each participant, we used the lights on/lights off times as sanity checks before proceeding with our analysis. We have modified the sentence in the original text (Line 133) as follows:

“The detection of the start of sleep is based on 10-minute segments and is totally independent of the sleep/wake scoring described earlier as the threshold is 3 activity counts in a 30s epoch. The process starts by looking at the first 10 minutes (20 30s-epochs) after lights-out time (which was synchronized to the PSG). Each epoch is tested against the threshold (i.e. 3 counts) and the number of epochs exceeding the threshold is counted. If this number is greater than 2 then the process is repeated 1 minute later. This process continues until a 10-minute block that fulfils the criteria is found marking the start of sleep.  Detecting the end of sleep is done using the same procedure, however, instead of 10-minute segments of maximum 2 epochs containing 3 or more activity counts, 5-minute segments of maximum 2 epochs containing 5 or more activity counts marks the end of sleep. Note that the lights out time was always synchronized to the start of the PSG recording.”

The Authors used Pearson correlations to assess the relationship between the sleep parameters of the devices and the PSG. However, Person correlation is not the best method for assessing agreement between different measures (see Bland, J. M., & Altman, D. G. (1999). Measuring agreement in method comparison studies. Statistical methods in medical research, 8(2), 135-160). Please consider to use more appropriate analyses, such as intraclass correlations, or to clearly state, for non-expert readers, that Pearson correlations cannot provide any information about the measurement error or of the agreement between devices. We thank the reviewer for this remark. We are aware of the fact that simple correlations do not provide any useful information about the agreement between two methods as correlation analysis mainly evaluates the linear association between two sets of observations (Bland and Altman 1999). However, we were careful not to over interpret such correlations. In fact, the only reason why we incorporated such basic correlations was to show that even at the very superficial level there is still no agreement between the measurements of the devices and that of the gold standard. That is, even before looking at the epoch-by-epoch agreement, simple average values over a night the PSG gold-standard and consumer device outputs show alarming disagreements. Please see in the discussion, Line 368:

“Although these correlations (see suppl. table S1) are not sufficient for commenting on the agreement between the sleep trackers and the PSG, they are important to show that even this simple relation does not hold statistically with alarming disagreements.”

Looking at the Bland-Altman plots, it seems to me that both the MB and the SC showed also a problem of heteroscedasticity (i.e., increase error as a function of the magnitude of the measured value) for SOL and WASO. The same problem, but only for SOL, seems to happen for the MW. Please consider to mention and discuss these results. We agree that this could be one possible interpretation of the results. However, there is a second interpretation that we believe fits much better to our results. That is, by simply observing the Bland and Altman plots of the WASO and SOL especially for the MB and the SC (see figure 3) we indeed notice and increase in the error with the increase in observed values of SOL and WASO. However, note that the increase in the WASO and SOL values here reflects deviations from the norm values expected for the general population (apprixmately 10min for WASO or SOL). For example, WASO values are generally around 10 minutes in normal healthy population and therefore a higher WASO than that represents a deviation from the norm (which the devices seem not to “expect”). Consequently, the increase in the error in that case might rather suggest that these devices/application take into account the expected norm values of the general population and perform really bad when the actual recorded night deviates from these norm values. Note that also our Sleep Efficiency (SE) results are in favor of our interpretation as here it is exactly the opposite case. More specifically, at lower SE, which is not normal in the healthy population (since about 90-95% SE is expected), there is an increase in the variance which further supports the argument that the consumer devices perform poorly when the recorded night deviates from the expected norm of the general population.

The authors used Cohen’s Kappa coefficient to assess the agreement between the devices, reporting very low values. Please consider that for sleep, Cohen’s Kappa is a very liberal index, since there is a higher proportion of sleep epochs compared to wake epochs. This can lead to ‘‘the first paradox of kappa statistic’’ (see Feinstein, A. R., & Cicchetti, D. V. (1990). High agreement but low kappa: I. The problems of two paradoxes. Journal of clinical epidemiology, 43(6), 543-549. and Cicchetti, D. V., & Feinstein, A. R. (1990). High agreement but low kappa: II. Resolving the paradoxes. Journal of clinical epidemiology, 43(6), 551-558.), that is, when two measures show high agreement but a low kappa. A more appropriate and conservative measure is the Prevalence- And Bias-Adjusted Kappa (PABAK), which gives balanced weight to sleep and wake epochs (see also Cellini, N., Buman, M. P., McDevitt, E. A., Ricker, A. A., & Mednick, S. C. (2013). Direct comparison of two actigraphy devices with polysomnographically recorded naps in healthy young adults. Chronobiology International, 30(5), 691-698.). Please consider reanalyzing the data with this approach or, as an alternative, to discuss that the results have been obtained with a low conservative approach. Thanks to the reviewer for this important critique, in the initial manuscript we report Cohen’s Kappa alongside the overall agreement. From the overall agreement one can simply calculate the PABAK and therefore have an indication on the influence of bias and prevalence on the reported agreement. Moreover, we mentioned in the discussion that such low Kappa in the presence of a high overall agreement might suggest a bias in the devices to score sleep (this is an important argument addressed in our discussion). We agree with the reviewer on using PABAK to circumvent this limitation and we have incorporated PABAK and the prevalence and bias indices in our results and respectively modified the discussion in the lights of the new results.

Discussion, Line 390:

“However, it is important to note that Cohen’s Kappa is affected by the imbalanced marginal totals in a table [15], masking high levels of agreement. Therefore, we report PABAK which has been shown to more accurate in such cases [16]. When sleep was scored into 3 categories (Wake/light sleep/deep sleep), both K and PABAK were very low (for the MB: k=0.14, PABAK=0.06 and for the SC: k=0.18, PABAK=-0.07), confirming the poor agreement .between the MB/SC and the PSG gold standard. Interestingly, when sleep was scored into two categories (Wake/Sleep), K scores dropped to half (MB: k=0.08 and SC: k=0.13), but PABAK increased at least twofold (MB: PABAK=0.72 and SC: PABAK=0. 30). This might indicate a bias, especially in the MB algorithm, which again raises the question whether such a biased output can be of any benefit to the user.” 

8. In the discussion, the authors say that “devices and applications had high accuracy in estimating the most global sleep parameter, namely TiB. This makes these devices a helpful tool for objectively measuring the time spent in bed at home rather than relying solely on subjective measures such as daily sleep diaries” (p.12, l.251-3). In my opinion, this sentence may be toned done, since, if I understood correctly, all the devices were turned on just before going to sleep in the same condition (lab) for most of the subjects, reducing the variability of TiB. To claim that TiB detection was accurate the authors may show the TiB detection in the subsample of participants who slept at home. That wm

We would like to draw the reviewer’s attention to the fact that even in the at-home recordings the participants were asked to turn on the device right before going to sleep which indeed reduces the variability of the TIB estimates recorded by the consumer devices. However, this only applies for the Sleep Cycle application as the Mi band is always recording and does not require manual initiation. We have modified the statement as suggested and added the TiB plots in the supplementary material (Figure S3).

Please consider to discuss the results in lights of the finding of a recent review of all the validated sleep trackers against PSG (de Zambotti, M., Cellini, N., Goldstone, A., Colrain, I. M., & Baker, F. C. (2019). Wearable Sleep Technology in Clinical and Research Settings. Medicine and science in sports and exercise, 51(7):1538-1557) We thank the reviewer for referring us to that very recent publication. We have discussed the the arguments in this paper in the manuscript on Line 441 as follows:

“However, the main advantage of the commercially available sleep trackers such as the MB and the SC in their current status is their unmatched affinity with the public which would encourage them to participate in research with a huge impact on the field of sleep research and sleep medicine. These sleep trackers can be useful means for collecting huge amounts of data that otherwise require a lot of time and money to collect and help us better understand sleep and tackle sleep disorders, given that they undergo the necessary validation steps [17,18].”

Reviewer 3 Report

 The manuscript is well written in scientific way and fluent English language. The article handles validation of commercial sleep trackers against gold-standard polysomnography (PSG). You made conclusion, which is quite obvious and already known. Succeeded recordings and analyzed nights are quite few. I also doubt using only somnolyzer automatic software in scoring of PSGs, at least double manual/human expert scoring can be taken account as comparison. This is not true validation against PSG but comparison to automatic scoring program.

I have few questions and concerns, which need to be answered:

In section materials and methods you describe the used protocol and device, but in this kind of journal (sensors) I would like to describe more detailed the operation principles of these tested devices. What biosignal is used for sleep scoring, movement or heart rate measurement etc.?

Laboratory PSGs were done with Geodesis 256-channel sensor net. How this affect sleep quality of test person. It certainly increase wakenings, arousals, and fragmented sleep and ruin quality of EEG-signal.

What this mean in line 102: the sensitivity threshold was set to 20 activity counts.

Line111 at p<.01, preferable p < 0.01

You wrote that REM epochs were excluded, so please express that also in results section. What this means in minutes in standard PSG TST?  This exclusion is understandable in your setup but it is hard to get to know what this mean in your results. Obvious TST values are smaller than in normal sleep. It is good to know how much your test persons sleep REM sleep with this setup.

Then at line 205 you take REM anyway account?  hard to read and understand.

Line 225 OA appear first time and expression can be found from line 231. This does not make reading easy.

Author Response

First of all, we would like to thank all reviewers for their time and effort as well as their interest in our work. The review process has substantially improved our manuscript and we hope that our responses are to the full satisfaction of the reviewers.

Reviewer 3

The manuscript is well written in scientific way and fluent English language. The article handles validation of commercial sleep trackers against gold-standard polysomnography (PSG). You made conclusion, which is quite obvious and already known. Succeeded recordings and analyzed nights are quite few. I also doubt using only somnolyzer automatic software in scoring of PSGs, at least double manual/human expert scoring can be taken account as comparison. This is not true validation against PSG but comparison to automatic scoring program.

I have few questions and concerns, which need to be answered:

In section materials and methods you describe the used protocol and device, but in this kind of journal (sensors) I would like to describe more detailed the operation principles of these tested devices. What biosignal is used for sleep scoring, movement or heart rate measurement etc.? We thank the reviewer for the feedback. We have received the same request from other reviewer as well. We have provided more information about the devices and applications in the methods section in a subsection “Instruments”, it reads as follows: . We addressed this issue in the methods section and a whole sub-section is added (page 3 Line 100) to describe the 3 devices /application used in more details. It reads as follows:

“Instruments: Mi-Band (MB; Xiaomi, Beijing, China): The Mi Band is a wrist-worn commercial activity tracker (15.7mm X10.5mmX 40.3mm, and weighs 7.0g) that detects sleep using a combination of two sensors; i) a proximity sensor and ii) actigraphy (actimetric sensor). While the proximity sensor detects contact with the skin, the actimetric sensor detects body movements to count sleep time and differentiate between light and deep. These data are then converted to a hypnogram using the Mi Fit software (algorithm v1.1.14,   Anhui Humai Infomration Techniology CO., Ltd., Hefei, China). We used the Mi band 2, however, some of the recordings were done using the Mi band 3 which, as advised by Xiaomi developers on their website, has no hardware update from the Mi band 2 that might influence the results. For our analysis, we used the hypnogram (graph) produced by the Mi Fit software to extract the exact timings of the start and the end of sleep, the start and the end of each sleep stage (light sleep/deep sleep) as well as the duration of the wake WASO in minutes. Using these data we calculated the TiB as the difference between the start and the end times reported in the software and the TST as the total amount of time spent asleep (both light or deep sleep) during this period. SOL is defined as the difference between the start time of the PSG recording and the time when the MB starts recording sleep “Light sleep”. Finally, SE was calculated as follows; SE= (TST/TiB)*100. The 30s epochs for the epoch-by-epoch agreement were extracted manually by fragmenting the segments of wake/light sleep/Deep sleep in the graph into 30s epochs.

MotionWatch8 (MW; CamNTech, Cambridge, UK):  The MW is a research-grade actigraphy with a built-in ambient light sensor (Dimesnions: 36mm l X 28.2 w X 9.4mm d excluding strap, and weighs 9.1 grams including battery but excluding strap; Sensor: Triaxial, MEMs technology, 0.01g to 8g range, 3 – 11Hz). For differentiating between sleep and wake epochs we used the MotionWare software (v1.1.20, empire Software GmbH, Cologne, Germany) which utilizes an algorithm that depends on thresholding. Briefly, this thresholding algorithm assigns an activity score to each epoch by totalizing the epoch in question and those surrounding it using weighting factors. If the activity score of an epoch is above a predefined threshold, then the epoch is scored as wake, otherwise it is scored as sleep. This activity score is dependent upon the sampling epoch-length. Since we used 30s epochs, the activity scores are calculated as follows: the number of movements in the epoch being scored is multiplied by 2 and that of the epochs within 2 minutes of this epoch (1 minute before and 1 minute after) are multiplied by 0.2. The activity score of this epoch is the sum of these weighted values. The MW has 3 thresholding options; low, medium and high corresponding to 80, 40, 20 activity score respectively. For our analysis we used the high threshold value as it was used for the validation of the MW. Moreover, an additional level of scoring is also done based on the movements detected by the MW per epoch. If there is more than 2 movements in a 30s epoch, this epoch is scored as mobile (awake). The detection of the start of sleep is based on 10-minute segments and is totally independent of the sleep/wake scoring described earlier as the threshold is 3 activity counts in a 30s epoch. The process starts by looking at the first 10 minutes (20 30s-epochs) after lights-out time (which was synchronized to the PSG). Each epoch is tested against the threshold (i.e. 3 counts) and the number of epochs exceeding the threshold is counted. If this number is greater than 2 then the process is repeated 1 minute later. This process continues until a 10-minute block that fulfils the criteria is found marking the start of sleep.  Detecting the end of sleep is done using the same procedure, however, instead of 10-minute segments of maximum 2 epochs containing 3 or more activity counts, 5-minute segments of maximum 2 epochs containing 5 or more activity counts marks the end of sleep. TiB, TST, SOL and SE are calculated automatically by the software and WASO was calculated manually. TiB is defined as the the total elapsed time between the “Lights Out” and “Got Up” times. TST is defined as the total elapsed time between the “Fell Asleep” and “Woke Up” times. SOL is defined as the time between “Lights Out” and “Fell Asleep” and SE as the “Actual Sleep Time” divided by “Time in Bed” in percentage. Finally, WASO was determined by calculating the amount of epochs scored as awake after the first epoch of sleep. 30s epoch data provided by the software was used to calculate the epoch-by-epoch agreement with gold standard.

Sleep Cycle (SC; NorthCube, Gothenberg, Sweden): SC is a mobile phone application that is available on android-based as well as IOS-based devices. SC is a smart alarm clock that tracks your sleep patterns and wakes you up during light sleep. SC tracks sleep throughout the night and use a 30-minute window that ends up with the desired alarm time during which the alarm goes off at the lightest possible sleep stage (i.e. light sleep). SC scores sleep through motion detection via one of two motion detection modes; i) microphone: which uses the built-in microphone is used to analyze movements or ii) accelerometer which uses the built-in accelerometer. SC tracks movements through the night and uses them to detect and score sleep as well as plotting a graph (hypnogram). For our analysis we used the SC on both IOS (v5.7.1) and Android (v3.0.1). We were advised by the developers (Northcube, GOT, Sweden) that there is no difference between the two versions in sleep scoring algorithms). We used the recommended settings for recording, that is, we used the microphone to detect movements and we placed the phone next to the participants with the microphone facing the participant and the charger plugged in. Although we used the premium version, we were not able to find any information on the algorithm by which SC is able to dissociate between sleep and wakefulness as well as to classify of sleep into light and deep sleep. As even the premium version of the application provides no access to the raw data we performed image analysis of the hypnogram provided by the application (see suppl. material and supplementary figure S1 for more information) as a workaround to this problem. Through this workaround we were able to extract 30s-epoch information on sleep scoring (Wake/Light sleep/Deep sleep) which was also used for measuring sleep parameters. TiB was calculated based on the “went to sleep” and woke up times reported by the application. TST was calculated by subtracting all the “awake” epochs from the TiB. SOL was calculated from by summing up all the awake 30s epochs before the first light sleep epoch. WASO was calculated by summing up all the awake epochs that lie between the first light sleep epoch and the “woke up” time. SE = (TST/TiB)*100.”

Laboratory PSGs were done with Geodesis 256-channel sensor net. How this affect sleep quality of test person. It certainly increase wakenings, arousals, and fragmented sleep and ruin quality of EEG-signal.

 By drawing a simple comparison between the data recorded using hdEEG in the laboratory and the ambulatory EEG system for the home recordings we were not able to find any statistical differences in the sleep parameters as depicted in Table S4 below (also added to the supplements). We do find a trend to shorter sleep onset latency at home as in lab as expected as due to the first night effect in the laboratory.

Moreover, regarding the quality of the signal, there shouldn’t be any effect on sleep scoring as our laboratory uses a fixed standard procedure for sleep scoring to ensure the quality and the reproducibility of the data and the analyses. That is, we always used the same set of electrodes for sleep staging no matter which device we are using ambulatory or in lab (that is, electrodes C3_A2, C4_A1, F3_A2, F4_A1, O1_A2, O2_A1, EOG and bipolar EMG). This setup is the recommended setup by the American Association for Sleep Medicine (AASM & Iber 2007) to ensure replicability across labs internationally. Please also note that for the agreement of our PSG gold standard with the consumer devices it should not matter at all whether there is more awakenings and/or fragmentation in sleep as the devices should be able to capture all of these patterns. In fact, this might be an advantage of such setup since we add more variance to sleep which would better mimic the sleep patterns across the population if we aim to discuss the benefits of such sleep trackers for the general public.

Reference:

American Academy of Sleep Medicine. (2007). The AASM manual for the scoring of sleep and associated events: rules, terminology and technical specifications. Westchester, IL: American Academy of Sleep Medicine, 23.

Table S4. A comparison between sleep parameters as measured by the laboratory hdEEG and the at-home ambulatory EEG systems.

Parameter

In-lab hdEEG (n=17)

Ambulatory EEG (n=8)

P*

TiB

452.29 ± 81.78  min

396.94 ± 117.54 min

0.26

TST

378.5 ± 95.21 min

352.31 ± 127 min

0.61

SOL

30 ± 20.49 min

17.44 ± 14.19 min

0.09+

WASO

44.44 ± 38.66 min

27.69 ± 37.80 min

0.32

SE

82.41 ± 13.83%

87.63 ± 11.89%

0.35

*Independent sample t-test without assuming equal variances (Welch approximation t-test). + p<0.1 or a statistical trend. TiB: time in bed; TST: total sleep time; SOL: sleep onset latency; WASO: wake after sleep onset; SE: sleep efficiency.

3. What this mean in line 102: the sensitivity threshold was set to 20 activity counts.

Thanks to the reviewer for drawing our attention to the shortage of information on the algorithm of the MotionWatch. The answer to this question has been added in details in the methods section “Instruments [page 3, paragraph 120] but we would also like to directly answer the reviewer here for convenience:

For differentiating between sleep and wake epochs we used the MotionWare software (v1.1.20, empire Software GmbH, Cologne, Germany) which utilizes an algorithm that depends on thresholding. That is, an activity score is given to each epoch by totalizing the epoch in question and those surrounding it using weighting factors based on epoch length. If the activity score of an epoch is above a predefined threshold, then the epoch is scored as wake, otherwise it is scored as sleep. The activity score is dependent upon the sampling epoch-length. Since we used 30s epochs, the activity scores are calculated as follows: the activity count of the epoch being scored multiplied by 2 and that of the epochs within 2 minutes of this epoch (1 minute before and 1 minute after) are multiplied by 0.2. The activity score of this epoch is the sum of these weighted values. The MW has 3 thresholding options; low, medium and high corresponding to 80, 40, 20 activity score respectively. For our analysis we used the high threshold value as it was used for the validation of the MW. This means that if the activity score is higher than 20, the epochs is scored as awake, if it is equal to or less than 20 then the epochs is scored as sleep.”

4. Line111 at p<.01, preferable p < 0.01

Thanks to the reviewer, this has been adjusted in the revised manuscript.

5. You wrote that REM epochs were excluded, so please express that also in results section. What this means in minutes in standard PSG TST?  This exclusion is understandable in your setup but it is hard to get to know what this mean in your results. Obvious TST values are smaller than in normal sleep. It is good to know how much your test persons sleep REM sleep with this setup.

We have clearly stated in the methods section that REM epochs were excluded only in the case of 3 category scoring (Wake/light sleep/deep sleep). That is, when we measured the agreement between the devices and the PSG in scoring sleep into 3 stages. However, when we measured the agreement in the case of 2-category scoring (i.e. the scoring into either Wake/Sleep) we included REM epochs as well (Table.2). Moreover, we have added a table in the supplementary material in which we did the agreement in the case of 2-category scoring (i.e. the scoring into either Wake/Sleep) while excluding REM (Supplementary material, table S2) and no difference in the agreement scores was found. we have modified the text in the original manuscript to be easier to read as follows:

Methods, Epochwise agreement section, Line 216:

“Importantly, when scoring sleep into 3 categories (Wake/Light sleep/Deep sleep) we excluded PSG epochs which were scored as stage REM according to the AASM from the analysis as all 3 devices and applications provide no information about REM (or “dreaming”) sleep. However, in the case of scoring sleep into 2 categories only (Wake/Sleep) REM epochs were included.”

Results section, Line 314

“Moreover, when we excluded REM epochs from this analysis, no significant difference in the agrrement scores (OA and K) was observed (see suppl. Table S2).”

To answer the reviewer specifically, we added a table in the supplementary material with the TST with and without REM (Table S6)

Table S6. The amount of time slept by each participant before and after removing the time spent in REM “dreaming” sleep.

TST with REM

TST without REM

383.5

317

81.5

81.5

432.5

373.5

423

375.5

335.5

305.5

502

438

438

386.5

381.5

302

307.5

259.5

504

441.5

337

301.5

420.5

389

317.5

266

386

307.5

391.5

329

362

303.5

431

353.5

425

316.5

390

295

419.5

364.5

508.5

455

205.5

190

398

331.5

120.5

112

351.5

310

6. Then at line 205 you take REM anyway account? hard to read and understand.

Thanks to the reviewer, this has been addressed in the original manuscript. We have clearly stated in the methods section that REM epochs were excluded only in the case of 3 category scoring (Wake/light sleep/deep sleep). That is, when we measured the agreement between the devices and the PSG in scoring sleep into 3 stages. However, when we measured the agreement in the case of 2-category scoring (i.e. the scoring into either Wake/Sleep) we included REM epochs as well (Table.2). Moreover, we have modified the text in the original manuscript to be easier to read as follows:

 “For the PSG gold standard light sleep was defined as stages N1 and N2 while Deep sleep is defined as N3 stages. Importantly, when scoring sleep into 3 categories (Wake/Light sleep/Deep sleep) we excluded PSG epochs which were scored as stage REM according to the AASM from the analysis as all 3 devices and applications provide no information about REM (or “dreaming”) sleep. However, in the case of scoring sleep into 2 categories (Wake/Sleep) REM epochs were included”.

7. Line 225 OA appear first time and expression can be found from line 231. This does not make reading easy.

Thanks to the reviewer, this has been adjusted in the revised manuscript. The abbreviation OA is mentioned after the first time the expression “overall agreement” is mentioned; Line 308

Reviewer 4 Report

The study investigated the accuracy of Mi band and actigraphy and reported the poor agreement with the PSG. This can be interesting to many readers. Please consider several comments. 1. Is there any difference between "laboratory" test and "home" test? In other words, I wonder if the accuracy of MB or SC was better at home. High-density-EEG with a 256-electrode at laboratory is excessive and may be uncomfortable for the participants compared to the 16-channel EEG. 2. In line 84-86, this study used computer-assisted sleep classification, not manual scoring. Actually, the SIESTA had low agreement with manual scoring (81-82%) (Neuropsychobiology. 2010;62(4):250-64.) The computer-assisted sleep scoring can be limitation of this study. 3. The SOL and WASO are a little high considering that the participants were healthy and relatively young (Sleep. 2004 Nov 1;27(7):1255-73.). I think these can contribute the low accuracy because the difference between consumer device and PSG is high when SE is high (= SOL and WASO are high). Please discuss this problem and limitation (possibly due to first-night effect in laboratory which can be overcome by serial night tests). 4. As the authors discussed in line 273, the MB underestimated SOL and WASO because "MB does not score awakenings from sleep unless they are almost unmistakable". I totally agree. However, I cannot estimate the real values from the presented data. If possible, please make a table with all possible data (TIB, TST, SE, SOL, WASO, Wake, light sleep, deep sleep) of each methods (PSG, MB, MW, SC). 5. I experienced that the trend match substantially between PSG and consumer device. I think drawing a duplicate hypnogram (PSG&MB or PSG&SC) of a well-matched participant or a poor-matched participant and discuss the cause of poor accuracy of this study. 6. In line 143, "supplementary figure S2-5", but there are no 3-5 figures. 7. In figure 1(A), I think x-axis is not "mean (min)", but "TIB of PSG (min)" If it is, revise all x-axis of figure 1-3.

Author Response

First of all, we would like to thank all reviewers for their time and effort as well as their interest in our work. The review process has substantially improved our manuscript and we hope that our responses are to the full satisfaction of the reviewers.

Reviewer 4

The study investigated the accuracy of Mi band and actigraphy and reported the poor agreement with the PSG. This can be interesting to many readers. Please consider several comments.

1. Is there any difference between "laboratory" test and "home" test? In other words, I wonder if the accuracy of MB or SC was better at home. High-density-EEG with a 256-electrode at laboratory is excessive and may be uncomfortable for the participants compared to the 16-channel EEG.

There shouldn’t be any differences in the signal between the two setups. By drawing a simple comparison between the data recorded using hdEEG in the laboratory and the ambulatory EEG system for the home recordings we were not able to find any statistical differences in the sleep parameters as depicted in Table S4 below (also added to the supplements). We do find a trend to shorter sleep onset latency at home as in lab as expected as due to the first night effect in the laboratory.

Moreover, regarding the quality of the signal, there shouldn’t be any effect on sleep scoring as our laboratory uses a fixed standard procedure for sleep scoring to ensure the quality and the reproducibility of the data and the analyses. That is, we always used the same set of electrodes for sleep staging no matter which device we are using ambulatory or in lab (that is, electrodes C3_A2, C4_A1, F3_A2, F4_A1, O1_A2, O2_A1, EOG and bipolar EMG). This setup is the recommended setup by the American Association for Sleep Medicine (AASM & Iber 2007) to ensure replicability across labs internationally. Please also note that for the agreement of our PSG gold standard with the consumer devices it should not matter at all whether there are more awakenings and/or fragmentation in sleep as the devices should be able to capture all of these patterns. In fact, this might be an advantage of such setup since we add more variance to sleep which would better assess the performance of such sleep trackers (mimicking the variance in the general public) if we aim to discuss their benefits for the general public.

Reference:

American Academy of Sleep Medicine. (2007). The AASM manual for the scoring of sleep and associated events: rules, terminology and technical specifications. Westchester, IL: American Academy of Sleep Medicine, 23.

Table S4. A comparison between sleep parameters as measured by the laboratory hdEEG and the at-home ambulatory EEG systems.

Parameter

In-lab hdEEG (n=17)

Ambulatory EEG (n=8)

P*

TiB

452.29 ± 81.78  min

396.94 ± 117.54 min

0.26

TST

378.5 ± 95.21 min

352.31 ± 127 min

0.61

SOL

30 ± 20.49 min

17.44 ± 14.19 min

0.09+

WASO

44.44 ± 38.66 min

27.69 ± 37.80 min

0.32

SE

82.41 ± 13.83%

87.63 ± 11.89%

0.35

*Independent sample t-test without assuming equal variances (Welch approximation t-test). + p<0.1 or a statistical trend. TiB: time in bed; TST: total sleep time; SOL: sleep onset latency; WASO: wake after sleep onset; SE: sleep efficiency.

2. In line 84-86, this study used computer-assisted sleep classification, not manual scoring. Actually, the SIESTA had low agreement with manual scoring (81-82%) (Neuropsychobiology. 2010;62(4):250-64.) The computer-assisted sleep scoring can be limitation of this study.

We have added in the supplementary material (Table S8), the agreement between a manual scorer and the 3 devices. The results are not different from those obtained when using the SIESTA computer-assisted scoring. The same low level agreement is observed for all devices and therefore the same conclusion would be drawn. Therefore, we thank the reviewer for this remark, however, we would like to disagree that the SIESTA semi-automated scoring is a limitation of this study since it produced the same results as the manual scoring. Also note that the agreement between the SOMNOLyzer and a manual scorer in scoring sleep was the same as, in fact slightly better than, the agreement between two manual scorers (Anderer et al., 2010). Therefore, we opted for keeping our results and not resort to the manual scoring as the gold standard.

Reference:

Anderer, P., Moreau, A., Woertz, M., Ross, M., Gruber, G., Parapatics, S., … Dorffner, G. (2010). Computer-assisted sleep classification according to the standard of the American Academy of Sleep Medicine: Validation study of the AASM version of the Somnolyzer 24 × 7. Neuropsychobiology

3. The SOL and WASO are a little high considering that the participants were healthy and relatively young (Sleep. 2004 Nov 1;27(7):1255-73.). I think these can contribute the low accuracy because the difference between consumer device and PSG is high when SE is high (= SOL and WASO are high). Please discuss this problem and limitation (possibly due to first-night effect in laboratory which can be overcome by serial night tests).

That is a very interesting point from the reviewer. We agree with the reviewer’s observation that the difference between the devices and the gold standard is high when SOL and WASO are high and the SE is low. In that case, however, having a high WASO and SOL is beneficial as it points out to a possible bias by the devices when they deviate from the normal values. Our hypothesis was that it shouldn’t matter how the values we get differ from the normal range as this should be captured by the devices as well. However, if the devices are biased in the sense that they always “expect” norm values they will fail to accurately estimate sleep when it deviates from the normal mean (norm). And this is exactly what the reviewer mentions and we observed in our data. Therefore, we wouldn’t consider having values that deviate from the normal range a limitation per se, but rather another dimension on which we could assess the performance of the devices. We also explicitly discuss this in the discussion line 370 as follows:

“This raises the question of whether the faulty estimation of values such as TST, SE, WASO or SOL are due to a priori knowledge of these sleep trackers of the amount of time the average person actually sleeps or needs to fall asleep. If such information is included in the algorithms and outputs of the consumer devices, this would explain why the largest errors occur primarily for “non-average” sleep profiles and nights. In line with this observation, previous studies have highlighted the poor performance of sleep trackers when sleep deviates from the average person’s sleep [6,8]. However, to date this argument remains speculative as all tested devices the MB and the SC do not allow raw data access or are black boxes when it comes to their staging algorithms. Similarly, when comparing the agreement between the MB and SC with the PSG gold standard for 3 sleep-wake classes (light sleep, deep sleep, and wake) as compared to 2 classes (sleep vs. wake) we found the expected increase in the OA yet a drop in the Kappa scores. Especially for the MB, looking at the sensitivity scores we observed extremely low sensitivity in detecting wakefulness (5.5%) and a very high sensitivity in detecting sleep (99.5%). That is, by assigning “sleep” to basically every epoch the device also cannot miss sleep epochs, yet it of course strongly overestimates sleep and has a vast amount of false alarms for stage “sleep”. Although the MB was the least sensitive between all the 3 devices and applications, it had the highest precision in scoring wakefulness (PPV: 62.8% for the MB, 47.8% for the MW and 24.3% for the SC). That is, the MB does not score awakenings from sleep unless they are almost unmistakable. This indicates a strong bias of the MB algorithm (as observed in the very low kappa values: 0.08; [10]), which again raises the question if such a biased output can be of any benefit to the end-user.”

As the authors discussed in line 273, the MB underestimated SOL and WASO because "MB does not score awakenings from sleep unless they are almost unmistakable". I totally agree. However, I cannot estimate the real values from the presented data. If possible, please make a table with all possible data (TIB, TST, SE, SOL, WASO, Wake, light sleep, deep sleep) of each methods (PSG, MB, MW, SC). We kindly refer the reviewer to the following table which contains the required information. Also added to the supplementary material (Table S6)

Table S5. A comparison between sleep paremeters as measured by the PSG gold standard and the 3 sleep trackers; Mi band, MotionWatch and Sleep cycle.

Parameter

PSG (25)

MB (21)

SC (12)

MW (12)

TiB (min)

434.58

±13.22

456.38

±89.79

412.17

±102.23

472.10

±41.16

TST(min)

370.12

±104.43

447.28

±87.11

260.75

±101.10

417.17

±35.81

SOL (min)

25.98

±19.35

11.24

±20.18

30

±18.70

17.10

±20.73

WASO

(min)

39.10

±38.42

9.14

±20.24

121.33

44.94

43.75

±24.86

SE (%)

84.10

±13.22

97.92

±4.53

60.29

±15.32

88.47

±5.19

SOL: Sleep onset latency, WASO: wake after sleep onset, SE: sleep efficiency and TST: total sleep time, TiB: time in bed, PSG: polysomnography, MB: Mi Band, SC: Sleep Cycle, MW: MotionWatch.

I experienced that the trend match substantially between PSG and consumer device. I think drawing a duplicate hypnogram (PSG&MB or PSG&SC) of a well-matched participant or a poor-matched participant and discuss the cause of poor accuracy of this study. We have included a hypnogram for the PSG, MB and SC in the supplementary material (see suppl. Fig. S4) and referred to it in the discussion (Line 399). The part in the discussion reads as follows:

“Regarding the other key parameters evaluated, our results raise serious doubts whether such consumer devices and applications can to-date provide any reliable information about sleep-related health issues. The MB and the SC had better performance in tracking sleep at night possibly due to their inability to capture the subtle sleep dynamics throughout the night (see suppl. figure S4). Especially the revealed misjudgment in estimating key features of sleep such as SE, SOL and WASO are worrisome as they are important diagnostic criteria for quantifying clinically relevant bad sleep and sleep disorders such as insomnia [1116]. On the contrary, by providing such inaccurate information these consumer devices might even run risk to contribute to worse sleep and life quality as end-users may be concerned by the sometimes negative output highlighting bad nights of sleep [45].”

Figure S4. Hypnograms produced by the PSG gold standard (red), the Mi band (MB, blue) and the Sleep cycle application (SC, green), for the same night. An example of an unusual night with bad agreement of the PSG scores with the MB and the SC scores. Note the inability of the MB and the SC to detect the transient changes in the sleep architecture throughout the night. Moreover, The MB and the SC were inaccurate in capturing wake after sleep onset (WASO) and the sleep onset latency (SOL), the reason why they tend to over-/under-estimate such parameters (Figure 3). Also note that the MB started staging sleep with a light-sleep epoch 23 minutes after the PSG started recording (the black dot), which means that the previous time the MB was not able to detect any sleep and did not perform any sleep scoring, another problem that might contribute to the poor agreement of the MB scoring with the gold standard. The time before that we hypothetically marked it as “awake”.

In line 143, "supplementary figure S2-5", but there are no 3-5 figures. 7. In figure 1(A), I think x-axis is not "mean (min)", but "TIB of PSG (min)" If it is, revise all x-axis of figure 1-3. The x-axis of the BA plots is always the mean values of both measurements. For more details please refer to Bland, J.M., Altman, D.G., 1999. Measuring agreement in method comparison studies. Stat Methods Med Res 8, 135–160. https://doi.org/10.1177/096228029900800204

Round 2

Reviewer 2 Report

The authors addressed all my concerns. I have no further comments.

Author Response

We would like to thank the reviewer for the time and effort as well as the interest in our work. The reviewer comments and suggestions have improved our manuscript substantially.

Reviewer 3 Report

I still feel that this is not real validation against gold standard PSG ( laboratory PSGs were done with 256-channel EEG cap and against somnolyzer automatic scoring software and only 19 healthy patients). Supplementary data  is not enough to made this better.

In section materials and methods you do not succeed describe the biosignal and technical background of those validated devices. In this kind of journal (sensors) they should be included. Please describe more detailed the operation principles of these tested devices. What biosignal is used for sleep scoring;body movement, heart rate measurement and via that Heart rate variability or  electrodermal activity so via galvanic skin response they maybe monitor autonomic nervous system or do they record via microphone the sounds from body movements?

.

Author Response

First of all, we would like to thank the reviewer and we hope that our additional responses clarify further issues of the reviewer.

The reviewer had some further concerns about our study that we would like to address here:

I still feel that this is not real validation against gold standard PSG ( laboratory PSGs were done with 256-channel EEG cap and against somnolyzer automatic scoring software and only 19 healthy patients). Supplementary data is not enough to made this better. We would like to draw the reviewer’s attention to the publication by Anderer and colleagues (2010) which showed that the agreement between the semi-automated SOMNOLyzer and a manual scorer in scoring sleep was the same and, in fact slightly better than, the agreement between two manual scorers. Moreover, we would like to refer the reviewer to the supplementary Table S8. In this table we show the agreement between a manual scorer and the 3 devices and it is evident that the results are very similar to those obtained when using the SIESTA computer-assisted scoring. The same low level agreement is observed for all devices and therefore the same conclusion would be drawn if we used sleep scoring by a manual scorer as the gold standard.

Please not that in the field of sleep research the Siesta Algorithm is accepted as standard as the agreement between human scorers is known to be similar if not worse. This is mainly the problem of sleep staging criteria which are rather roughly defined and only tuned to 30 sec epochs to date. Please also note that we publish in prestigious high rank Journals since more than 10 years and always refer to that sleep staging algorithm as it allows the reproducibility of sleep-staging results over the years; and which is not the case if a human scorer is changing all the time.

Reference:

Anderer, P., Moreau, A., Woertz, M., Ross, M., Gruber, G., Parapatics, S., Dorffner, G. (2010). Computer-assisted sleep classification according to the standard of the American Academy of Sleep Medicine: Validation study of the AASM version of the Somnolyzer 24 × 7. Neuropsychobiology, 62(4), 250-64. doi: https://doi.org/10.1159/000320864.

In section materials and methods you do not succeed describe the biosignal and technical background of those validated devices. In this kind of journal (sensors) they should be included. Please describe more detailed the operation principles of these tested devices. What biosignal is used for sleep scoring;body movement, heart rate measurement and via that Heart rate variability or electrodermal activity so via galvanic skin response they maybe monitor autonomic nervous system or do they record via microphone the sounds from body movements? We thank the reviewer for that important feedback. We would like to also add the fact that the manufacturers of the consumer devices used for sleep tracking have usually not published any details on their devices and sensors or algorithms. It is therefore extremely difficult to collect detailed information about the hardware and software used by these sleep trackers as they remain “black boxes” which is one of the main criticisms of our respective publication. However, we have added more (and as much as possible) information to the method section regarding the tested devices:

Methods section; Line 105

The MB uses a tri-axial accelerometer and a photoplethysmography (PPG) sensor to detect movements and monitor blood volume changes, respectively. The Mi Band (MB) starts recording sleep when it detects no movement for a certain period of time. The MB classifies sleep into deep and light sleep based on body movements. In addition, the MB uses PPG for continuous heart rate monitoring during sleep to track light and deep sleep more precisely. However, information on the thresholds used for classifying sleep is not made available”.

Methods section; Line 127

“For scoring sleep, the MW uses a tri-axial micro-electro-mechanical system (MEMS) accelerometer (range: 0.01g-8g, bandwith: 3-11Hz) to monitor body movements with a sampling frequency of 50Hz. The on-board software processes the raw acceleration data such that one quantitative measure of activity for a pre-defined epoch length of e.g. 30s is calculated and stored on an internal non-volatile memory.”

Methods section; Line 176

“The Sleep Cycle (SC) application utilizes sound analysis to identify sleep phases by tracking movements in bed. The SC application uses the smartphone built-in microphones to pick up sounds from the sleeper. After receiving the sound input the application then filters the sound using a series of high and low cut-off filters to identify the specific noises that correlates with movement. When there is no motion the application registers deep sleep, when there is little motion it registers light sleep and when there is a lot of motion it registers wakefulness. More details on the algorithms and the technical aspects of sound analysis are not available to the public”

Reviewer 4 Report

The authors responded appropriately to the comments.

Author Response

(The authors gave the same response as above.)
